# Beyond Generative Priors: Minority Sampling with JEPA-Guided Diffusion

**Sol Park** [1]   **Soobin Um** [† 2]

## Abstract

Minority sampling aims to generate low-density instances on a data manifold and is of central importance in applications such as medical diagnosis, anomaly detection, and creative AI. Existing approaches, however, define minority samples relative to generative priors learned from training data, confining rarity to model-specific notions that may poorly reflect real-world semantics. In this work, we propose a world-centric perspective on minority sampling, which defines rarity with respect to real-world priors rather than generator-induced densities. To this end, we introduce *JEPA guidance*, a diffusion sampling framework guided by a Joint-Embedding Predictive Architecture (JEPA)—a class of world models that encode broad, semantically rich representations. JEPA guidance steers diffusion trajectories toward low-density regions under the implicit density induced by the JEPA, thereby aligning generated minorities with real-world semantic rarity. To make JEPA guidance computationally practical, we develop principled approximation strategies accompanied by theoretical error bounds, significantly reducing the overhead of guidance computation. Extensive experiments across unconditional, class-conditional, and text-to-image generation demonstrate that JEPA guidance consistently improves the fidelity and semantic validity of minority samples, outperforming generator-centric baselines in capturing real-world notions of rarity. Code is available at https://github.com/soobin-um/jepa-guidance.

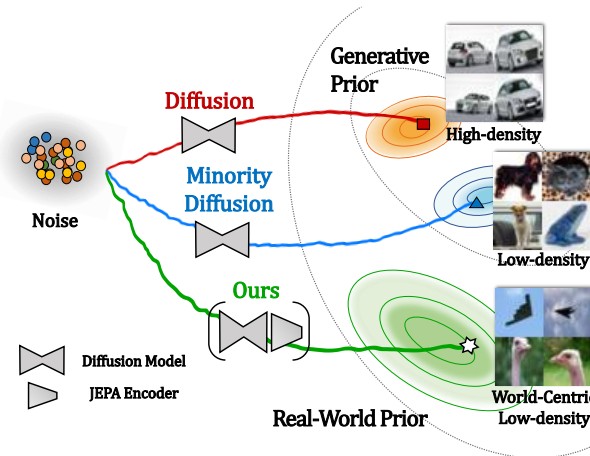

*Figure 1.* **Beyond generative priors: world-centric minority sampling.** Existing minority-guidance methods (blue) target low-density regions *within* the learned generative prior (Um et al., 2025), producing samples that are rare only under a specific training distribution (*e.g.*, a dog on a white background). Our approach (green) leverages a JEPA encoder—a promising candidate for world models (LeCun, 2022)—to guide diffusion toward low-density regions under a *world prior*, generating samples that are genuinely rare in the real world (*e.g.*, stealth aircraft).

## 1. Introduction

*Minority samples* refer to underrepresented instances that reside in low-density regions of the data manifold (Yu et al., 2020; Sehwag et al., 2022). These samples exhibit distinctive characteristics rarely observed in majority (high-density) regions, making them valuable for applications such as medical diagnosis (Um et al., 2024), anomaly detection (Du et al., 2023), and creative AI (Rombach et al., 2022; Han et al., 2022). However, synthesizing such underrepresented samples—known as *minority sampling*—remains challenging, as generative models tend to favor high-density regions (Sehwag et al., 2022). Recent advances in diffusion models (Song & Ermon, 2019; Ho et al., 2020) have opened new possibilities, with their strong capacity to model complex data distributions enabling substantial progress in this area (Um et al., 2024; Um & Ye, 2024; 2025; Um et al., 2025).

Despite these advances, existing minority samplers are fundamentally constrained by the prior implicitly defined by the generative model itself (Um et al., 2024; Um & Ye, 2024; 2025; Um et al., 2025). They identify minority samples only

†Corresponding author. [1]Agency for Defense Development, Daejeon, South Korea [2]Department of Artificial Intelligence, Kookmin University, Seoul, South Korea. Correspondence to: Soobin Um <soobin.um@kookmin.ac.kr>.

*Proceedings of the 43$^{rd}$ International Conference on Machine Learning*, Seoul, South Korea. PMLR 306, 2026. Copyright 2026 by the author(s).

relative to the model's own learned density, which merely captures a specific training set, leading to fragmented and model-specific notions of minority (Figure 1). This limitation becomes critical when considering minority definitions grounded in broader, real-world semantics. In particular, current approaches cannot generate minority samples with respect to priors captured by world models (Ha & Schmidhuber, 2018; LeCun, 2022)—an increasingly central component in the pursuit of artificial general intelligence.

In this work, we move beyond generative priors and introduce a new paradigm: *world-centric* minority sampling aligned with real-world priors. Our approach, *JEPA Guidance*, is a guided sampler that generates minority instances with respect to the implicit prior of Joint-Embedding Predictive Architectures (JEPAs) (LeCun, 2022; Assran et al., 2023). JEPAs are a promising class of world models (LeCun, 2022) whose learned representations have been shown to encode data density, providing a natural proxy for real-world priors.

Specifically at inference time, we first estimate the densities of intermediate latent samples induced by an off-the-shelf JEPA encoder (*e.g.*, DINOv2 (Oquab et al., 2023)). We then guide the diffusion sampling process toward low-density regions under this JEPA-induced prior, thereby encouraging the generation of samples that are rare with respect to the world prior rather than the generator's own density. However, we observe that naive optimization of JEPA-based densities using existing formulations (Balestriero et al., 2025) is computationally prohibitive during sampling. To address this, we incorporate efficient approximation techniques based on random matrix theory (Halko et al., 2011) with a theoretical upper bound, substantially reducing computational overhead. To further enhance practical applicability, we provide techniques to address the domain gap between the diffusion model and the JEPA encoder, along with an extension to conditional generation tasks.

Our extensive experiments demonstrate that JEPA guidance substantially improves the ability of diffusion models to produce high-fidelity minority samples aligned with real-world priors. We show that JEPA guidance is broadly applicable across a wide range of generative settings, including unconditional and class-conditional image generation, text-to-image models (*e.g.*, Stable Diffusion (Rombach et al., 2022)), and even few-step models such as SDXL-Lightning (Lin et al., 2024). To further demonstrate practical applicability, we explore a downstream application: data-augmented classification. Importantly, JEPA guidance is entirely training-free and can be readily integrated at inference time using only pretrained diffusion and JEPA models. Moreover, our approach is condition-agnostic: the JEPA encoder does not require access to conditioning information used by the diffusion model (*e.g.*, class labels or text

prompts), allowing JEPA guidance to be applied without modifying either model.

Our contributions are summarized as follows:

- We introduce a new perspective on minority sampling based on atypicality defined by real-world priors, rather than generative priors.
- We propose *JEPA guidance*, a diffusion-based method that steers sampling toward low-density regions under JEPA-induced priors.
- We empirically demonstrate that JEPA guidance generates high-fidelity minority samples that are atypical under world priors.

## 2. Related Work

**Minority Sampling with Diffusion Models.** The task of generating minority samples has been widely explored within the diffusion modeling framework, exploiting its capacity to access low-density regions of the data distribution. Early efforts in this direction predominantly relied on classifier-guided diffusion (Dhariwal & Nichol, 2021), in which auxiliary classifiers provide gradient-based signals to steer the sampling process toward minority regions (Sehwag et al., 2022; Um et al., 2024). Subsequent works have aimed to reduce the dependence on external classifiers by introducing self-contained minority guidance that can be computed solely using a pretrained diffusion model (Um & Ye, 2024). This line of research has enabled minority sample generation across diverse settings, including text-to-image synthesis (Um & Ye, 2025). More recently, the approach in Um et al. (2025) further improves practicality by proposing a guidance-free initialization strategy that encourages minority representations with only marginal additional computational cost. However, these methods remain fundamentally limited to defining minorities with respect to the generative model's own prior, and therefore cannot produce minority samples under broader real-world priors, such as those induced by world models (Ha & Schmidhuber, 2018). In contrast, our approach explicitly decouples the notion of minority from the generative prior by leveraging a world prior encoded in a JEPA, enabling minority sampling that is defined with respect to real-world representations rather than the generator itself.

**Joint-Embedding Predictive Architectures (JEPAs).** JEPAs are a family of self-supervised learning methods that learn by predicting in a joint embedding space (LeCun, 2022; Assran et al., 2023). The key idea is to encode both input $x$ and target $y$ into a shared embedding space, then train a predictor to map from $x$'s embedding to $y$'s embedding. This joint embedding-space prediction encourages learning semantic, high-level features rather than low-level details, making JEPAs a compelling candidate for modeling

real-world priors (LeCun, 2022). In practice, I-JEPA applies this principle to images by predicting the representations of masked target blocks from a context block (Assran et al., 2023), while V-JEPA extends it to videos for capturing temporally consistent structures (Assran et al., 2025). Among JEPA-family models, DINOs (Oquab et al., 2023; Siméoni et al., 2025) stand out for their scale (*e.g.*, trained on 142M diverse images for DINOv2 (Oquab et al., 2023)) and strong performance across various downstream tasks. Recent work further shows that JEPAs implicitly encode the probability density of their training data through their learned representations, introducing *JEPA-SCORE* (Balestriero et al., 2025)—a density estimator defined via the Jacobian of a JEPA encoder. This finding highlights the potential of JEPAs as a source of real-world priors for applications such as outlier detection and data curation. In this work, we move beyond using this density signal solely for post-hoc ranking. Instead, we integrate it as guidance during diffusion sampling, reframing minority generation from being defined by the generative model's narrow prior to being grounded in the broader, real-world representations captured by JEPAs.

## 3. Background

### 3.1. Diffusion-Based Generative Models

Diffusion models are a class of generative frameworks characterized by two paired processes: a forward diffusion process and a reverse denoising process. The forward process is an iterative noise-perturbation procedure that gradually corrupts a clean data point $x_0 \in \mathbb{R}^n$ into a fully noisy sample $x_T \in \mathbb{R}^n$. It is commonly defined as a sequence of Gaussian transitions parameterized by a variance schedule $\{\beta_t\}_{t=1}^T$:

$$q(x_t|x_{t-1}) = \mathcal{N}(x_t; \sqrt{1-\beta_t}x_{t-1}, \beta_t I), \quad (1)$$

where $\{x_t\}_{t=1}^{T-1}$ denote intermediate latent variables that share the same dimensionality as $x_0$. Notably, the forward process admits a closed-form marginal, $q(x_t|x_0) = \mathcal{N}(\sqrt{\alpha_t}x_0, (1-\alpha_t)I)$, where $\alpha_t := \prod_{s=1}^t (1-\beta_s)$. The reverse process corresponds to a progressive denoising procedure modeled by learnable Gaussian transitions:

$$p_\theta(x_{t-1}|x_t) := \mathcal{N}(x_{t-1}; \mu_\theta(x_t, t), \Sigma_\theta(x_t, t)), \quad (2)$$

where the mean and covariance are parameterized by a neural network with parameters $\theta$. Training diffusion models amounts to learning this network, typically by optimizing a variational bound (Ho et al., 2020) or an equivalent denoising score-matching objective (Song et al., 2020b). In particular, the DDPM training (Ho et al., 2020) yields the mean parameterized with a noise prediction network:

$$\mu_\theta(x_t, t) = \frac{1}{\sqrt{1-\beta_t}}\left(x_t - \frac{\beta_t}{\sqrt{1-\alpha_t}}\epsilon_\theta(x_t, t)\right),$$

where $\epsilon_\theta(x_t, t)$ is trained to predict noise added accordance to Eq. (1). The variance is often fixed, *e.g.*, $\Sigma_\theta(x_t, t) = \beta_t I$ (Ho et al., 2020).

Once trained, data generation can be done by iteratively sampling from the reverse transitions in Eq. (2), starting from $x_T \sim \mathcal{N}(0, I)$ and progressing toward $x_0$:

$$x_{t-1} = \mu_\theta(x_t, t) + \Sigma_\theta(x_t, t)^{1/2}z, \quad z \sim \mathcal{N}(0, I). \quad (3)$$

**Guided Sampling.** One instrumental feature of diffusion sampling is its amenability to post-hoc conditioning, commonly referred to as *guidance* (Song et al., 2020b). Specifically, at each timestep $t$, an arbitrary energy-based gradient can be incorporated into Eq. (3) to encourage the sampling process to progress toward a desired direction (Epstein et al., 2023):

$$x_{t-1} = \mu_\theta(x_t, t) + \Sigma_\theta(x_t, t)^{1/2}z + \eta_t g(x_t, t), \quad (4)$$

where $g(x_t, t)$ is a user-defined guidance function, and $\eta_t$ denotes the (time-dependent) controller that modulates the strength of guidance.

### 3.2. JEPA-SCORE: Implicit Density Learned by JEPAs

JEPA-SCORE is a density measure implicitly encoded by the learned representations of Joint-Embedding Predictive Architectures (JEPAs) (Balestriero et al., 2025). Given a pretrained JEPA encoder $f : \mathbb{R}^n \to \mathbb{R}^d$ (like DINOv2 (Oquab et al., 2023)), JEPA-SCORE is defined as the sum of the logarithms of the singular values of its Jacobian:

$$\mathrm{JS}(x) := \sum_{i=1}^r \log(\sigma_i(J_f(x))), \quad (5)$$

where $J_f(x) \in \mathbb{R}^{d \times n}$ denotes the Jacobian of $f$ at $x$, $\sigma_i(\cdot)$ denotes the $i$-th singular value, and $r := \mathrm{rank}(J_f(x))$.

Intuitively, JEPA-SCORE summarizes the local geometric effect of the JEPA encoder $f$ around the input $x$. The singular values of the Jacobian quantify how infinitesimal perturbations in the input space are amplified along orthogonal directions in the representation space. Prior work shows that JEPA-SCORE correlates with the underlying data density, assigning higher values to samples from denser regions and lower values to those from sparser regions of the data distribution (Balestriero et al., 2025).

In practice, computing JEPA-SCORE requires performing singular value decomposition (SVD) of the Jacobian. However, the Jacobian $J_f$ is often prohibitively large in realistic settings, making exact SVD computationally expensive. As a result, directly applying JEPA-SCORE in downstream tasks, particularly within iterative procedures such as diffusion sampling, becomes impractical.

# 4. Method

## 4.1. Redefining Minorities: from Generator-Centric to World-Centric

Our approach starts by reformulating the definition of minority samples, clarifying the limitations of conventional generator-centric perspectives. Minority samples are commonly defined as instances that reside in low-density regions of a data distribution. More formally, given a generative model with implicit density $p_{\boldsymbol{\theta}}$, minorities can be characterized as (Um et al., 2025)

$$\mathcal{S}_{\boldsymbol{\theta},\epsilon} := \{\boldsymbol{x} \in \mathcal{M}_{\boldsymbol{\theta}} : p_{\boldsymbol{\theta}}(\boldsymbol{x}) < \epsilon\}, \quad (6)$$

where $\mathcal{M}_{\boldsymbol{\theta}}$ denotes the data manifold induced by the generative model (*i.e.*, the support of $p_{\boldsymbol{\theta}}$), and $\epsilon$ is a small positive threshold. Existing minority sampling methods are explicitly designed to draw samples from Eq. (6) (Sehwag et al., 2022; Um et al., 2024; Um & Ye, 2024), thereby defining minorities solely with respect to the generative model's learned distribution. We refer to this formulation as *generator-centric* minority sampling.

However, such a definition inherently ties the notion of minority samples to the training data of the generative model, which may be limited in scale and thus fail to faithfully reflect real-world priors. Moreover, it is susceptible to the inductive biases and modeling limitations of the generative model itself. As a result, samples that are rare under the generative prior do not necessarily correspond to genuinely rare or atypical instances in the real world; see Figure 2a for instance.

To address this limitation, we propose to redefine minorities with respect to a world prior that is independent of the generative model. Specifically, we consider a representation-based prior induced by a pretrained JEPA, and define minority samples as those residing in low-density regions under this world-level representation (LeCun, 2022). Let $f_{\boldsymbol{\phi}}$ denote a JEPA network parameterized by $\boldsymbol{\phi}$. In this setting, minority samples are formally defined as

$$\mathcal{S}_{\boldsymbol{\phi},\epsilon} := \{\boldsymbol{x} \in \mathcal{M}_{\boldsymbol{\phi}} : p_{\boldsymbol{\phi}}(\boldsymbol{x}) < \epsilon\}, \quad (7)$$

where $p_{\boldsymbol{\phi}}$ denotes the implicit density induced by the JEPA representations, and $\mathcal{M}_{\boldsymbol{\phi}}$ is the manifold induced by the JEPA representation. Our objective is to generate samples from Eq. (7), thereby performing *world-centric* minority sampling.

Figure 2 exhibits an illustrative comparison of the two definitions on CIFAR-10 (Krizhevsky et al., 2009). As we can see, generator-centric minorities (defined by AvgkNN distance in the test set) are semantically dispersed across diverse classes, whereas world-centric minorities (defined by JEPA-SCORE) concentrate on specific semantic categories—notably ostriches and stealth aircraft, atypical instances

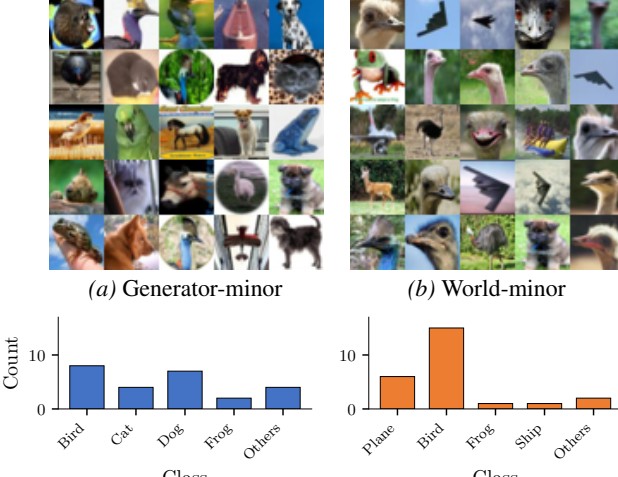

*(a)* Generator-minor      *(b)* World-minor

*Figure 2.* **Comparison of minority samples under generator-centric (a) and world-centric (b) definitions on CIFAR-10.** For each definition, we show the top-25 minority samples (top) and their class distribution (bottom). Generator-centric minorities, defined by AvgkNN distance in the test set, are semantically dispersed across classes. In contrast, world-centric minorities (via JEPA-SCORE with DINOv2 (Oquab et al., 2023)) concentrate on specific categories—notably birds (60%) and planes (24%), many of which are ostriches and stealth aircraft, atypical instances often underrepresented in the world prior captured by web-scale pretraining data of the JEPA. See Figure 4 for more comparisons.

often underrepresented in web-scale pretraining data for the JEPA encoder (Oquab et al., 2023). This observation suggests that world-centric minorities capture semantically meaningful atypicality, rather than mere statistical outliers in the training distribution. We provide an extended qualitative comparison of the two definitions on ImageNet $256 \times 256$ in Section B.1.

## 4.2. JEPA Guidance: World-Centric Minority Sampling

We now introduce JEPA Guidance, a principled mechanism for implementing world-centric minority sampling with diffusion models. The key idea is to incorporate a JEPA encoder $f_{\boldsymbol{\phi}}$ to guide sampling toward low-density regions under the JEPA-induced prior $p_{\boldsymbol{\phi}}$. A naive realization of this idea is to directly compute JEPA-SCORE (defined in Eq. (5)) at each sampling step and use its gradient as a guidance signal. However, as discussed in Section 3.2, computing JEPA-SCORE requires performing SVD on a large Jacobian, which is computationally prohibitive in practice.

**Approximation via Randomized SVD.** To address this challenge, we leverage randomized SVD (Halko et al., 2011) to efficiently approximate JEPA-SCORE. Let the SVD of the Jacobian of the JEPA encoder be

$$J_f(\boldsymbol{x}) = \boldsymbol{U}\boldsymbol{\Sigma}\boldsymbol{V}^{\top} \in \mathbb{R}^{d \times n},$$

where $\boldsymbol{\Sigma}$ contains the singular values of $J_f(\boldsymbol{x})$. Randomized SVD constructs an orthonormal projection matrix

$\boldsymbol{Q} \in \mathbb{R}^{d \times l}$ ($l \ll d$) such that $J_f(\boldsymbol{x}) \approx \boldsymbol{Q}\boldsymbol{Q}^\top J_f(\boldsymbol{x})$ (Halko et al., 2011). Using this projection, we define a compressed Jacobian

$$\tilde{J}_f(\boldsymbol{x}) := \boldsymbol{Q}^\top J_f(\boldsymbol{x}) \in \mathbb{R}^{l \times n},$$

whose SVD can be computed efficiently:

$$\tilde{J}_f(\boldsymbol{x}) = \tilde{\boldsymbol{U}}\tilde{\boldsymbol{\Sigma}}\tilde{\boldsymbol{V}}^\top.$$

We then approximate JEPA-SCORE using the top-$k$ singular values of the compressed Jacobian:

$$\bar{\text{JS}}(\boldsymbol{x}) := \sum_{i=1}^{k} \log\left(\tilde{\sigma}_i\big(\tilde{J}_f(\boldsymbol{x})\big)\right), \qquad (8)$$

where $\{\tilde{\sigma}_i\}_{i=1}^{k}$ denote the leading singular values of $\tilde{J}_f(\boldsymbol{x})$. In practice, we set $k = l - p$, where $p$ is an oversampling parameter used in randomized SVD to improve approximation quality (Halko et al., 2011). When combined with $q$ steps of power iteration, randomized SVD can accurately recover the leading singular subspace (Halko et al., 2011), yielding $\tilde{\sigma}_i \approx \sigma_i$ for the top-$k$ components. Below we provide a theoretical bound on our approximation error; see Section A for the proof.

**Proposition 4.1.** *The approximation error of Eq. (8) against Eq. (5) is upper bounded as:*

$$JS(\boldsymbol{x}) - \bar{JS}(\boldsymbol{x}) \leq \mathcal{E}_{RSVD}(\boldsymbol{x}) + \mathcal{E}_{Trunc}(\boldsymbol{x}), \qquad (9)$$

*where $\mathcal{E}_{RSVD}(\boldsymbol{x})$ denotes the error bound due to randomized SVD:*

$$\mathcal{E}_{RSVD}(\boldsymbol{x}) := \sum_{i=1}^{k} \log\left(1 + C_{k,q}\frac{\sigma_{k+1}\big(J_f(\boldsymbol{x})\big)}{\tilde{\sigma}_i\big(\bar{J}_{f,k}(\boldsymbol{x})\big)}\right), \quad (10)$$

*and $\mathcal{E}_{Trunc}(\boldsymbol{x})$ is the truncation error bound from disregarding $\{\sigma_i\}_{i=k+1}^{r}$:*

$$\mathcal{E}_{Trunc}(\boldsymbol{x}) := \frac{r-k}{2}\log\left(\frac{\left\|J_f(\boldsymbol{x}) - \bar{J}_{f,k}(\boldsymbol{x})\right\|_F^2}{r-k}\right). \quad (11)$$

*Here $\bar{J}_{f,k}(\boldsymbol{x})$ is the rank-$k$ approximation of $J_f(\boldsymbol{x})$ via randomized SVD, and $C_{k,q} := 1 + (1 + 4\sqrt{2r/(k-1)})^{1/(2q+1)}$ is a constant depending on rank $k$ and power iterations $q$ (Halko et al., 2011).*

Notice that the overall approximation error decomposes into two terms: the estimation error from randomized SVD and the truncation error from discarding $\{\sigma_i\}_{i=k+1}^{r}$. This implies that even when the top-$k$ singular values are accurately estimated, the approximation error can remain large if $k$ is chosen too small. However, we empirically find that $k \approx 10$ suffices for effective guidance, as the lower singular values

**Algorithm 1** JEPA-guided minority sampling

1: **Input:** $\epsilon_{\boldsymbol{\theta}}, f_{\boldsymbol{\phi}}, T, N, \tau, k, p, q, \eta_t$
2: $\boldsymbol{x}_T \sim \mathcal{N}(\boldsymbol{0}, \boldsymbol{I})$
3: **for** $t = T, T-1, \ldots, 1$ **do**
4: $\quad \boldsymbol{z} \sim \mathcal{N}(\boldsymbol{0}, \boldsymbol{I})$ if $t > 1$, else $\boldsymbol{z} = \boldsymbol{0}$
5: $\quad \boldsymbol{x}_{t-1} \leftarrow \boldsymbol{\mu}_{\boldsymbol{\theta}}(\boldsymbol{x}_t, t) + \boldsymbol{\Sigma}_{\boldsymbol{\theta}}^{1/2}(\boldsymbol{x}_t, t)\boldsymbol{z}$
6: $\quad$ **if** $t < \tau T$ **and** $t \bmod N = 0$ **then**
7: $\qquad \hat{\boldsymbol{x}}_{0|t} \leftarrow (\boldsymbol{x}_t - \sqrt{1 - \alpha_t}\epsilon_{\boldsymbol{\theta}}(\boldsymbol{x}_t, t))/\sqrt{\alpha_t}$
8: $\qquad J_f \leftarrow \nabla_{\hat{\boldsymbol{x}}_{0|t}} f_{\boldsymbol{\phi}}(\hat{\boldsymbol{x}}_{0|t})$
9: $\qquad \boldsymbol{Q}^* \leftarrow \text{RANDSVD}(J_f, k, p, q)$
10: $\qquad \text{JS}^* \leftarrow \sum_{i=1}^{k} \log\big(\tilde{\sigma}_i(\text{sg}(\boldsymbol{Q}^{*\top})J_f)\big)$
11: $\qquad \boldsymbol{x}_{t-1} \leftarrow \boldsymbol{x}_{t-1} - \eta_t \nabla_{\boldsymbol{x}_t}\text{JS}^*$
12: $\quad$ **end if**
13: **end for**
14: **Return** $\boldsymbol{x}_0$

could act as a near-constant offset across samples, contributing little to variations in atypicality. We provide an in-depth analysis in Section B.2.

**Guidance with the Envelope Theorem.** Our next step is to construct a guidance function that optimizes Eq. (8) during diffusion sampling. Since the JEPA encoder $f_{\boldsymbol{\phi}}$ operates on clean samples $\boldsymbol{x}_0$, not the noisy latents $\boldsymbol{x}_t$ encountered during inference, we evaluate it on a denoised estimate $\hat{\boldsymbol{x}}_{0|t}$. This yields the following guidance term:

$$\tilde{\boldsymbol{g}}(\boldsymbol{x}_t, t) := -\nabla_{\boldsymbol{x}_t}\bar{\text{JS}}(\hat{\boldsymbol{x}}_{0|t}),$$

where $\hat{\boldsymbol{x}}_{0|t} := \big(\boldsymbol{x}_t - \sqrt{1 - \alpha_t}\,\epsilon_{\boldsymbol{\theta}}(\boldsymbol{x}_t, t)\big)/\sqrt{\alpha_t}$ is the denoised estimate of $\boldsymbol{x}_0$ given $\boldsymbol{x}_t$ (Chung et al., 2023). However, this formulation suffers from a critical limitation: the objective in Eq. (8) retains a computational graph with respect to $\boldsymbol{x}_t$ through both $\boldsymbol{Q}$ and $\tilde{J}_f$, leading to excessive memory consumption and computational overhead. In particular, $\boldsymbol{Q}$ is obtained via an inner optimization that depends on $J_f(\hat{\boldsymbol{x}}_{0|t})$ (Halko et al., 2011), and thus implicitly on $\boldsymbol{x}_t$.

To address this issue, we invoke the Envelope Theorem (Milgrom & Segal, 2002). At the optimum of the inner randomized SVD problem, the theorem guarantees that the gradient of the outer objective $\bar{\text{JS}}(\hat{\boldsymbol{x}}_{0|t})$ with respect to $\boldsymbol{x}_t$ can be computed by treating the optimal projection $\boldsymbol{Q}^*$ as a constant. As a result, the dependence of $\boldsymbol{Q}^*$ on $\boldsymbol{x}_t$ can be safely ignored when differentiating Eq. (8):

$$\nabla_{\boldsymbol{x}_t}\bar{\text{JS}}(\hat{\boldsymbol{x}}_{0|t}) = \nabla_{\boldsymbol{x}_t}\text{JS}^*(\hat{\boldsymbol{x}}_{0|t}),$$

where JS$^*$ denotes the JEPA-SCORE evaluated with the optimal projection treated as a constant:

$$\text{JS}^*(\hat{\boldsymbol{x}}_{0|t}) := \sum_{i=1}^{k} \log\big(\tilde{\sigma}_i\big(\text{sg}(\boldsymbol{Q}^{*\top})J_f(\hat{\boldsymbol{x}}_{0|t})\big)\big),$$

and $\text{sg}(\cdot)$ denotes the stop-gradient operator (Chen & He, 2021). This formulation eliminates backpropagation

| DDIM | MinorityPrompt | Ours | DDIM | MinorityPrompt | Ours |
|---|---|---|---|---|---|

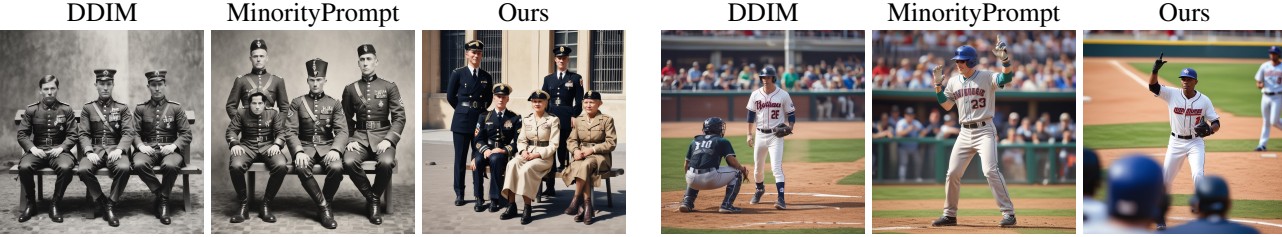

"Three persons with military attire seated on a bench"    "A baseball player on the home plate during a game"

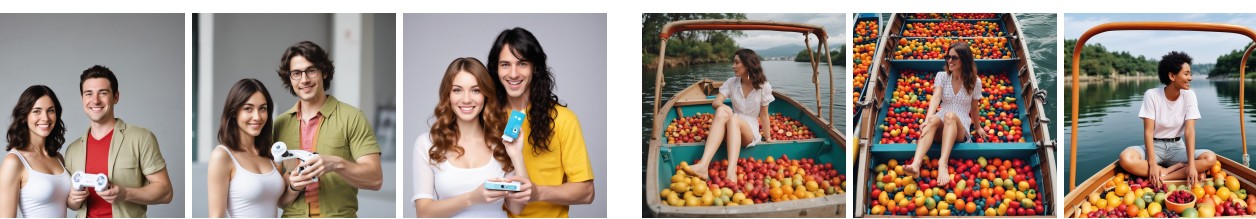

"A beautiful woman next to a man with a Wii controller"    "A woman sitting in a boat filled with fruit"

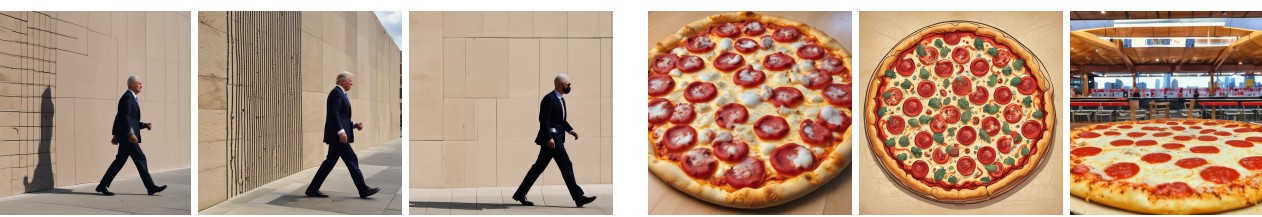

"The president of the US walking by a wall"    "This large pizza has a lot of cheese and tomato sauce"

*Figure 3.* **Sample comparison on SDXL-Lightning.** Generated samples from three approaches: (i) DDIM (Song et al., 2020a), (ii) MinorityPrompt (Um & Ye, 2025), and (iii) Ours. Six prompts were used, and random seeds were shared across all methods.

through the randomized SVD procedure, substantially reducing memory usage and computational overhead while preserving correct first-order gradients. The proposed JEPA guidance is then defined as:

$$\boldsymbol{g}^*(\boldsymbol{x}_t, t) := -\nabla_{\boldsymbol{x}_t} \mathrm{JS}^*(\hat{\boldsymbol{x}}_{0|t}). \tag{12}$$

From an optimization perspective, JEPA guidance can be interpreted as a bilevel problem: the inner level estimates a representation-level density via randomized SVD, while the outer level guides diffusion sampling toward low-density regions.

**Practical Considerations.** To further reduce computational overhead, we apply JEPA guidance intermittently, *e.g.*, every $N$ sampling steps, following Um & Ye (2024). For the guidance step size, we use either a variance-scaled schedule $\eta_t = \eta \boldsymbol{\Sigma}_{\boldsymbol{\theta}}(\boldsymbol{x}_t, t)$ or a constant $\eta_t = \eta$. Implementation details are provided in Sections C and 5, and a computational analysis in Section B.3.

### 4.3. Addressing the Domain Gap in JEPA Guidance

Although JEPA guidance uses denoised estimates $\hat{\boldsymbol{x}}_{0|t}$ to bridge the gap between the diffusion model and the JEPA encoder, we found this alone to be insufficient. Applying Eq. (12) throughout the entire sampling process yields limited performance gains, which we attribute to the domain

gap between the clean inputs $\boldsymbol{x}_0$ expected by $f_{\boldsymbol{\phi}}$ and the denoised estimates $\hat{\boldsymbol{x}}_{0|t}$—particularly in early timesteps where $\hat{\boldsymbol{x}}_{0|t}$ remains noisy and blurry.

To address this, we propose *deferred guidance*, which delays the start of JEPA guidance until an intermediate timestep $\tau T$. For instance, with $\tau = 0.8$, we begin JEPA guidance at $t = 0.8T$ rather than $t = T$, avoiding blurry denoised images during early sampling steps. We found this simple strategy to be highly effective; see Table 3b for empirical evidence.

### 4.4. Extension to Conditional Diffusion Models

Deferred guidance offers an additional benefit: it enables JEPA guidance to extend to conditional generation, one of the most successful applications of modern diffusion models (Rombach et al., 2022). The JEPA encoder $f_{\boldsymbol{\phi}}$ is inherently condition-agnostic, *i.e.*, it cannot incorporate the conditioning information used by conditional diffusion models. As such, it is naturally unable to guide toward low-density regions while respecting a given condition, *e.g.*, minimizing $p_{\boldsymbol{\phi}}(\boldsymbol{x}_0|\boldsymbol{c})$.

However, deferred guidance naturally addresses this limitation by allowing the conditional diffusion model to sample unguided from $t = T$ to an intermediate timestep (*e.g.*, $t = 0.8T$). By this point, the samples have already incorpo-

| CelebA 64×64 | cFID ↓ | sFID ↓ | Prec ↑ | Rec ↑ | Den ↑ | Cov ↑ | JEPA ↓ |
|---|---|---|---|---|---|---|---|
| ADM | 12.11 | 6.35 | **0.85** | 0.57 | **1.28** | 0.97 | -221.67 |
| Sehwag et al. | 61.61 | 18.21 | 0.63 | 0.70 | 0.40 | 0.49 | -138.71 |
| MG | 58.27 | 16.56 | 0.71 | 0.67 | 0.49 | 0.37 | -145.90 |
| SGMS | 61.76 | 20.42 | 0.62 | **0.84** | 0.38 | 0.49 | -171.85 |
| BnS | 67.10 | 15.65 | 0.55 | 0.83 | 0.35 | 0.72 | -202.89 |
| Ours | **8.50** | **4.94** | 0.82 | 0.65 | 1.09 | **0.98** | **-300.79** |

| ImageNet 256×256 | cFID ↓ | sFID ↓ | Prec ↑ | Rec ↑ | Den ↑ | Cov ↑ | JEPA ↓ |
|---|---|---|---|---|---|---|---|
| ADM | 26.44 | 9.70 | **0.95** | 0.51 | **1.52** | 0.96 | -102.01 |
| Sehwag et al. | 42.33 | 10.39 | 0.93 | 0.48 | 1.42 | 0.92 | -71.82 |
| MG | 34.84 | 10.42 | 0.93 | 0.53 | 1.32 | 0.95 | -118.76 |
| SGMS | 37.90 | 10.76 | 0.91 | 0.58 | 1.15 | 0.94 | -114.94 |
| BnS | 32.01 | 10.61 | 0.92 | 0.56 | 1.22 | 0.96 | -125.77 |
| Ours | 18.33 | 7.62 | 0.92 | **0.68** | 1.15 | **0.99** | **-241.62** |

*Table 1.* **Quantitative comparison on unconditional and class-conditional generation.** Prec/Rec denote Improved Precision/Recall (Kynkäänniemi et al., 2019); Den/Cov denote Density/Coverage (Naeem et al., 2020). JEPA denotes JEPA-SCORE (lower = more atypical). Best in **bold**, second best underlined.

| SDv1.5 | CLIP ↑ | Pick ↑ | ImgR ↑ | Prec ↑ | Rec ↑ | Den ↑ | Cov ↑ | JEPA ↓ |
|---|---|---|---|---|---|---|---|---|
| DDIM | 31.52 | 21.49 | 0.21 | **0.71** | 0.72 | **0.64** | **0.77** | -292.27 |
| CADS | 31.46 | 21.28 | 0.09 | 0.68 | 0.74 | 0.56 | 0.74 | -311.23 |
| SGMS | 31.23 | 21.32 | 0.12 | 0.65 | 0.70 | 0.48 | 0.69 | -311.14 |
| MinorityPrompt | **31.56** | 21.32 | **0.24** | 0.67 | 0.73 | 0.52 | 0.73 | -322.33 |
| Ours | 31.46 | **21.50** | 0.22 | 0.67 | **0.75** | 0.53 | 0.72 | **-355.40** |

| SDXL-Lightning | CLIP ↑ | Pick ↑ | ImgR ↑ | Prec ↑ | Rec ↑ | Den ↑ | Cov ↑ | JEPA ↓ |
|---|---|---|---|---|---|---|---|---|
| DDIM | **31.57** | 22.68 | **0.73** | 0.67 | 0.67 | 0.51 | 0.66 | -283.04 |
| CADS | 31.08 | 22.37 | 0.49 | **0.67** | 0.67 | **0.53** | 0.65 | -276.60 |
| SGMS | 31.36 | 22.58 | 0.68 | 0.58 | **0.77** | 0.38 | 0.58 | -318.03 |
| MinorityPrompt | 31.36 | 22.62 | 0.71 | 0.59 | 0.72 | 0.39 | 0.57 | -302.17 |
| Ours | 31.52 | **22.63** | **0.73** | 0.62 | 0.72 | 0.45 | 0.63 | **-337.88** |

*Table 2.* **Quantitative comparison on text-to-image generation.** CLIP, Pick, ImgR denote CLIPScore, PickScore, ImageReward. Prec/Rec denote Improved Precision/Recall (Kynkäänniemi et al., 2019); Den/Cov denote Density/Coverage (Naeem et al., 2020). JEPA denotes JEPA-SCORE (lower = more atypical). Best in **bold**, second best underlined.

rated substantial conditioning information. Consequently, JEPA guidance in the later stages steers samples toward low-density regions while preserving the conditional structure established earlier. This enables world-centric minority sampling in various conditional settings, including class-conditional (Dhariwal & Nichol, 2021) and text-to-image generation (Rombach et al., 2022), without modifying the JEPA encoder; see Tables 1 and 2. We summarize JEPA guidance in Algorithm 1.

# 5. Experiments

**Datasets and Pretrained Models.** We conduct experiments on three generative tasks: (i) unconditional, (ii) class-conditional, and (iii) text-conditional generation. For unconditional generation, we use CelebA $64 \times 64$ (Liu et al., 2015) with the checkpoint from Um & Ye (2024). For class-conditional generation, we use ImageNet $256 \times 256$ (Deng et al., 2009) with pretrained models from Dhariwal & Nichol (2021). For text-conditional generation, we use Stable Diffusion 1.5 (Rombach et al., 2022) and SDXL-Lightning (4-step) (Lin et al., 2024), evaluated on MS-COCO prompts (Lin et al., 2014).

**Baselines.** We compare against various samplers, with a focus on minority sampling approaches. For CelebA and ImageNet, we consider four minority samplers: (i) Sehwag et al. (2022), (ii) MG (Um et al., 2024), (iii) SGMS (Um & Ye, 2024), and (iv) BnS (Um et al., 2025). For text-conditional generation, we consider three minority (or diversity) methods: (i) CADS (Sadat et al., 2023), (ii) SGMS (Um & Ye, 2024), and (iii) MinorityPrompt (Um & Ye, 2025). We also include standard samplers such as ADM (Dhariwal & Nichol, 2021) and DDIM (Song et al., 2020a).

**Evaluation Metrics.** We employ various metrics to evaluate quality and diversity of generated samples. For

CelebA and ImageNet, we use: (i) Clean Fréchet Inception Distance (cFID) (Parmar et al., 2022), (ii) Spatial FID (sFID) (Nash et al., 2021), (iii) Improved Precision & Recall (Kynkäänniemi et al., 2019), and (iv) Density & Coverage (Naeem et al., 2020). To evaluate closeness to world-centric minorities, we use samples with the lowest JEPA-SCORE—the most atypical under the world prior—as reference data for computing these metrics, similar to Um et al. (2024); Um & Ye (2024); Um et al. (2025). For text-conditional generation, we use text-alignment metrics[1]: (i) CLIPScore (Hessel et al., 2021), (ii) PickScore (Kirstain et al., 2023), and (iii) ImageReward (Xu et al., 2023). For all settings, we use JEPA-SCORE (in Eq. (5)) to measure the degree of atypicality. We further conduct a human preference study for text-conditional generation; see Section D.1.

## 5.1. Results

**Qualitative Comparisons.** Figure 3 compares generated samples from our approach against two baselines on SDXL-Lightning. While MinorityPrompt (Um & Ye, 2025) generates samples with low-density features (*e.g.*, complex textures or patterns) compared to standard samplers, JEPA guidance produces more globally atypical semantics, such as elderly female military personnel, African-American baseball players, and long-haired males (see the first and second rows). Additional samples are provided in Section D.2; see Figures 6 to 9 therein.

**Quantitative Results.** Table 1 shows the performance of JEPA guidance on unconditional and class-conditional generation. Our approach outperforms all baselines in generating high-fidelity world-centric minority samples across both tasks. Notably, existing minority samplers such as

---

[1]We do not include FID for text-conditional generation, as it evaluates reproduction of MS-COCO images, which diverges from our focus on low-density generation.

| $\eta$ | CLIP↑ | Pick↑ | ImgR↑ | JEPA↓ |
|---|---|---|---|---|
| 0.00 | **31.44** | 21.46 | 0.19 | -289.79 |
| 0.02 | 31.43 | **21.47** | 0.20 | -307.33 |
| 0.06 | 31.36 | **21.47** | **0.20** | **-344.76** |

*(a) Impact of guidance strength $\eta$*

| $\tau$ | CLIP↑ | Pick↑ | ImgR↑ | JEPA↓ |
|---|---|---|---|---|
| 1.0 | 31.26 | 21.33 | 0.11 | -356.22 |
| 0.9 | 31.31 | 21.42 | 0.20 | -356.72 |
| 0.8 | **31.40** | 21.46 | **0.23** | **-360.82** |

*(b) Influence of deferring ratio $\tau$*

| $k$ | CLIP↑ | Pick↑ | ImgR↑ | JEPA↓ |
|---|---|---|---|---|
| 3 | **31.56** | **22.59** | 0.72 | -325.35 |
| 9 | 31.52 | **22.59** | 0.70 | **-344.85** |
| 15 | 31.53 | 22.58 | **0.73** | -335.28 |

*(c) Effect of RSVD rank $k$*

| **DINOv2-ViT-S/14** | CLIP↑ | Pick↑ | ImgR↑ | JEPA↓ |
|---|---|---|---|---|
| 0.00 | **31.44** | 21.46 | 0.19 | -289.79 |
| 0.04 | 31.43 | **21.47** | 0.21 | -324.59 |
| 0.08 | 31.37 | **21.47** | **0.22** | **-361.52** |

| **DINOv2-ViT-L/14** | CLIP↑ | Pick↑ | ImgR↑ | JEPA↓ |
|---|---|---|---|---|
| 0.00 | 31.44 | 21.46 | 0.19 | -1650.16 |
| 0.04 | **31.45** | **21.47** | **0.21** | -1687.41 |
| 0.08 | 31.41 | 21.46 | **0.21** | **-1739.36** |

| **MetaCLIP** | CLIP↑ | Pick↑ | ImgR↑ | JEPA↓ |
|---|---|---|---|---|
| 0.00 | **31.44** | **21.46** | **0.19** | -722.89 |
| 0.04 | 31.40 | **21.46** | 0.18 | -749.07 |
| 0.08 | 31.37 | 21.45 | 0.17 | **-776.52** |

*(d) Impact of JEPA encoder $f_\phi$*

*Table 3.* **Exploring the design space of JEPA guidance.** We study the impact of (a) guidance step size $\eta$, (b) deferral ratio $\tau$, (c) rank $k$ for randomized SVD, and (d) choice of JEPA encoder. CLIP, Pick, ImgR denote CLIPScore, PickScore, ImageReward. JEPA denotes JEPA-SCORE (lower = more atypical).

SGMS (Um & Ye, 2024) struggle to produce world-centric minorities with low JEPA-SCORE values, even underperforming standard sampling methods like ADM (Dhariwal & Nichol, 2021). This suggests that world-centric minority sampling cannot be achieved by simply applying existing generator-centric approaches. Table 2 presents results on text-to-image generation. JEPA guidance achieves the lowest JEPA-SCORE while maintaining competitive text-alignment scores, demonstrating its effectiveness in this challenging setting. On SDv1.5, MinorityPrompt exhibits higher CLIPScore and ImageReward but lower PickScore. We find that PickScore better captures quality issues such as noise and artifacts that CLIPScore and ImageReward often miss (see Figure 8). Our approach remains effective with few-step models such as SDXL-Lightning, highlighting its practical applicability.

**Ablation Studies.** Table 3 investigates the impact of key design choices in JEPA guidance. As expected, increasing the step size $\eta$ encourages greater atypicality in generated samples, with some trade-off in text-alignment. We also observe in Table 3b that deferred guidance is crucial: without it ($\tau = 1.0$), both quality and text-alignment degrade significantly. Table 3c shows that increasing the rank $k$ of randomized SVD is generally beneficial, but with diminishing returns—improvements between $k = 9$ and $k = 15$ are marginal. We provide an analysis to support this phenomenon; see Section B.2. Table 3d demonstrates the performance across different JEPA encoders. JEPA guidance consistently improves the generation of world-centric atypical samples regardless of encoder choice, demonstrating its robustness.

**Downstream Application.** To demonstrate the practical utility of JEPA guidance, we investigate its application in classifier training with synthetic data augmentation. Following Um & Ye (2024); Um et al. (2025), we train ResNet-18 models to predict 40 attributes in CelebA using five configurations: (i) CelebA training set only; (ii)–(iv) augmented with 50K samples from ADM (Dhariwal & Nichol, 2021), SGMS (Um & Ye, 2024), and BnS (Um et al., 2025), respec-

| Training data | Acc ↑ | F1 ↑ | Prec ↑ | Rec ↑ | #Aug ↓ |
|---|---|---|---|---|---|
| CelebA trainset | 0.898 | 0.746 | 0.815 | 0.710 | – |
| + ADM samples | 0.897 | 0.742 | 0.808 | 0.711 | 50K |
| + SGMS samples | **0.903** | 0.757 | **0.822** | 0.724 | 50K |
| + BnS samples | 0.902 | 0.755 | 0.819 | 0.723 | 50K |
| + Ours samples | 0.902 | **0.775** | **0.824** | **0.731** | **30K** |

*Table 4.* **Data augmentation for downstream classification.** "Trainset" denotes training on the CelebA training set only. Other rows indicate augmentation with samples from each method. "#Aug" denotes the number of augmented samples. All results are evaluated on the CelebA testset and averaged over three runs.

tively; and (v) augmented with 30K samples from ours. Generated samples are labeled using an off-the-shelf classifier. As shown in Table 4, augmenting with our world-centric minority samples yields the best or comparable performance despite using fewer samples, suggesting that atypical samples under the world prior provide more informative training signals than generator-centric minorities.

## 6. Conclusion

We introduced a new perspective on minority sampling by redefining minorities with respect to world priors rather than generator-centric priors. To realize this, we proposed JEPA guidance, which steers diffusion sampling toward low-density regions under JEPA-induced priors. Our approach leverages randomized SVD and the envelope theorem for efficient computation, and employs deferred guidance to bridge the domain gap and enable conditional generation. Experiments demonstrate that JEPA guidance consistently generates high-fidelity world-centric minority samples across various settings.

**Limitations and Future Work.** While JEPA guidance is effective, it introduces additional computational overhead from Jacobian computation and randomized SVD at each guidance step. Exploring more efficient approximations or amortized approaches could further improve scalability. Additionally, investigating other world models beyond JEPA encoders, or combining multiple priors, may lead to richer notions of atypicality.

## Impact Statement

This paper presents a framework for generating atypical samples guided by world priors. We believe this work can benefit applications such as data augmentation, robustness testing, and creative content generation. It may also help improve fairness and inclusivity in downstream tasks that rely on generated samples.

However, we note a potential dual-use concern: by reversing the guidance direction (*i.e.*, using $-\eta$ instead of $\eta$), our framework could be repurposed to generate high-density samples that reinforce existing biases in the world prior, potentially amplifying demographic or stylistic homogeneity. We encourage practitioners to use JEPA guidance responsibly and to consider the societal implications of generated content.

## Acknowledgments

This work was supported by the Agency for Defense Development grant funded by the Korean Government (No. 912A45701), the National Research Foundation of Korea (NRF) grant funded by the Korean Government (MSIT) (No. RS-2026-25541956), and the Institute of Information & Communications Technology Planning & Evaluation (IITP) grant funded by the Korean Government (MSIT) (No. RS-2025-02219317, AI Star Fellowship (Kookmin University)).

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

# A. Proof of Proposition 4.1

**Proposition 4.1.** *The approximation error of Eq. (8) against Eq. (5) is upper bounded as:*

$$JS(\boldsymbol{x}) - \bar{JS}(\boldsymbol{x}) \leq \mathcal{E}_{RSVD}(\boldsymbol{x}) + \mathcal{E}_{Trunc}(\boldsymbol{x}), \tag{13}$$

*where $\mathcal{E}_{RSVD}(\boldsymbol{x})$ denotes the error bound due to randomized SVD:*

$$\mathcal{E}_{RSVD}(\boldsymbol{x}) := \sum_{i=1}^{k} \log\left(1 + C_{k,q} \frac{\sigma_{k+1}\left(J_f(\boldsymbol{x})\right)}{\tilde{\sigma}_i\left(\bar{J}_{f,k}(\boldsymbol{x})\right)}\right), \tag{14}$$

*and $\mathcal{E}_{Trunc}(\boldsymbol{x})$ is the truncation error bound from disregarding $\{\sigma_i\}_{i=k+1}^{r}$:*

$$\mathcal{E}_{Trunc}(\boldsymbol{x}) := \frac{r-k}{2} \log\left(\frac{\left\|J_f(\boldsymbol{x}) - \bar{J}_{f,k}(\boldsymbol{x})\right\|_F^2}{r-k}\right). \tag{15}$$

*Here $\bar{J}_{f,k}(\boldsymbol{x})$ is the rank-k approximation of $J_f(\boldsymbol{x})$ via randomized SVD, and $C_{k,q} := 1 + (1 + 4\sqrt{2r/(k-1)})^{1/(2q+1)}$ is a constant depending on rank $k$ and power iterations $q$ (Halko et al., 2011).*

*Proof.* We start by manipulating the LHS of Eq. (13):

$$\begin{aligned} JS(\boldsymbol{x}) - \bar{JS}(\boldsymbol{x}) &= \sum_{i=1}^{r} \log\left(\sigma_i(J_f(\boldsymbol{x}))\right) - \sum_{i=1}^{k} \log\left(\tilde{\sigma}_i\left(\bar{J}_f(\boldsymbol{x})\right)\right) \\ &= \underbrace{\sum_{i=1}^{k} \log\left(\frac{\sigma_i(J_f(\boldsymbol{x}))}{\tilde{\sigma}_i\left(\bar{J}_{f,k}(\boldsymbol{x})\right)}\right)}_{\text{Randomized SVD error}} + \underbrace{\sum_{i=k+1}^{r} \log\left(\sigma_i\left(J_f(\boldsymbol{x})\right)\right)}_{\text{Truncation error}}. \end{aligned} \tag{16}$$

We first show that the error term for randomized SVD is upper bounded by $\mathcal{E}_{\text{RSVD}}(\boldsymbol{x})$ in Eq. (14). To this end, we start from the spectral-norm error bound of the randomized SVD approximation (Halko et al., 2011):

$$\left\|J_f - \bar{J}_{f,k}\right\| \leq C_{k,q}\, \sigma_{k+1}\left(J_f\right), \tag{17}$$

where $C_{k,q} := 1 + (1 + 4\sqrt{2r/(k-1)})^{1/(2q+1)}$. We drop the notation on $\boldsymbol{x}$ for simplicity.

For each $i \leq k$, we have:

$$\frac{\sigma_i(J_f)}{\tilde{\sigma}_i(\bar{J}_{f,k})} = 1 + \frac{\sigma_i(J_f) - \tilde{\sigma}_i(\bar{J}_{f,k})}{\tilde{\sigma}_i(\bar{J}_{f,k})} \leq 1 + \frac{\left|\sigma_i(J_f) - \tilde{\sigma}_i(\bar{J}_{f,k})\right|}{\tilde{\sigma}_i(\bar{J}_{f,k})}.$$

Taking the log of both sides gives:

$$\log\left(\frac{\sigma_i(J_f)}{\tilde{\sigma}_i(\bar{J}_{f,k})}\right) \leq \log\left(1 + \frac{\left|\sigma_i(J_f) - \tilde{\sigma}_i(\bar{J}_{f,k})\right|}{\tilde{\sigma}_i(\bar{J}_{f,k})}\right). \tag{18}$$

Here we use the singular value perturbation inequality (*i.e.*, Weyl's inequality (Lewis, 2019)):

$$\left|\sigma_i(J_f) - \tilde{\sigma}_i(\bar{J}_{f,k})\right| \leq \left\|J_f - \bar{J}_{f,k}\right\|.$$

Combining this inequality with Eq. (17) yields:

$$\left|\sigma_i(J) - \tilde{\sigma}_i(\bar{J}_{f,k})\right| \leq C_{k,q}\, \sigma_{k+1}(J_f).$$

Now plugging this into Eq. (18) and taking the summation over $i = 1, \ldots, k$ gives:

$$\underbrace{\sum_{i=1}^{k} \log\left(\frac{\sigma_i(J_f)}{\tilde{\sigma}_i(\bar{J}_{f,k})}\right)}_{\text{Randomized SVD error}} \leq \underbrace{\sum_{i=1}^{k} \log\left(1 + C_{k,q} \frac{\sigma_{k+1}(J_f)}{\tilde{\sigma}_i(\bar{J}_{f,k})}\right)}_{\mathcal{E}_{\text{RSVD}}(\boldsymbol{x})}. \tag{19}$$

Now we prove that the truncation error term is bounded by $\mathcal{E}_{\text{Trunc}}(\boldsymbol{x})$ in Eq. (15). We first obtain a simple upper bound for the truncation error via the Jensen's inequality:

$$\sum_{i=k+1}^{r} \log\left(\sigma_i(J_f)\right) = \frac{1}{2} \sum_{i=k+1}^{r} \log\left(\sigma_i^2(J_f)\right) \leq \frac{t}{2} \log\left(\frac{1}{t} \sum_{i=k+1}^{r} \sigma_i^2(J_f)\right). \tag{20}$$

By the Eckart–Young–Mirsky theorem (Damle, 2019) , we have

$$\sum_{i=k+1}^{r} \sigma_i^2(J_f) = \left\| J_f - J_{f,k} \right\|_F^2 \tag{21}$$

where $J_{f,k}$ denotes the rank-$k$ truncated version of $J_f$ using the *full* SVD. Since $\bar{J}_{f,k}$ is the approximation via randomized SVD, we have:

$$\left\| J_f - J_{f,k} \right\|_F^2 \leq \left\| J_f - \bar{J}_{f,k} \right\|_F^2. \tag{22}$$

Plugging Eq. (22) into Eq. (21) yields:

$$\sum_{i=k+1}^{r} \sigma_i^2(J_f) \leq \left\| J_f - \bar{J}_{f,k} \right\|_F^2.$$

Substituting into Eq. (20) gives:

$$\underbrace{\sum_{i=k+1}^{r} \log\left(\sigma_i(J_f)\right)}_{\text{Truncation error}} \leq \underbrace{\frac{r-k}{2} \log\left(\frac{\left\| J_f - \bar{J}_{f,k} \right\|_F^2}{r-k}\right)}_{\mathcal{E}_{\text{Trunc}}(\boldsymbol{x})}. \tag{23}$$

Plugging Eq. (19) and Eq. (23) into Eq. (16) and restoring the dependence on $\boldsymbol{x}$ yields:

$$\begin{aligned}
\text{JS}(\boldsymbol{x}) - \bar{\text{JS}}(\boldsymbol{x}) &= \sum_{i=1}^{k} \log\left(\frac{\sigma_i(J_f(\boldsymbol{x}))}{\tilde{\sigma}_i(\bar{J}_{f,k}(\boldsymbol{x}))}\right) + \sum_{i=k+1}^{r} \log\left(\sigma_i\left(J_f(\boldsymbol{x})\right)\right) \\
&\leq \sum_{i=1}^{k} \log\left(1 + C_{k,q} \frac{\sigma_{k+1}(J_f(\boldsymbol{x}))}{\tilde{\sigma}_i(\bar{J}_{f,k}(\boldsymbol{x}))}\right) + \frac{r-k}{2} \log\left(\frac{\left\| J_f(\boldsymbol{x}) - \bar{J}_{f,k}(\boldsymbol{x}) \right\|_F^2}{r-k}\right) \\
&= \mathcal{E}_{\text{RSVD}}(\boldsymbol{x}) + \mathcal{E}_{\text{Trunc}}(\boldsymbol{x}).
\end{aligned}$$

This completes the proof. $\qquad\square$

## B. Additional Analyses and Discussions

### B.1. Extended Comparison of Generator-Centric and World-Centric Minorities

Figure 4 provides a qualitative comparison of minority samples identified under the generator-centric and world-centric definitions on ImageNet $256 \times 256$. We visualize minority samples from four representative classes: bald eagle, white wolf, lemon, and volcano. Since ImageNet is a large-scale dataset, there exists some correlation between the two definitions, resulting in partially overlapping sets of minority samples (Balestriero et al., 2025). However, a closer examination reveals a qualitative distinction in what each definition captures.

The generator-centric definition tends to identify samples with atypical *context*—for instance, an eagle flying near a construction crane, wolves in unusual environmental settings, or lemons with cluttered backgrounds—highlighting outliers that lie far from the majority of training samples in pixel or feature space. In contrast, the world-centric definition more frequently identifies samples exhibiting atypical *behavior or appearance* of the object itself, such as mating eagles, wolves with distinctive facial expressions, unusually shaped lemons, or volcanoes captured from rare perspectives. This distinction highlights that, grounded in real-world priors, the world-centric definition captures semantically meaningful atypicality by focusing on the rarity of the object itself, rather than mere distributional outliers tied to the training data.

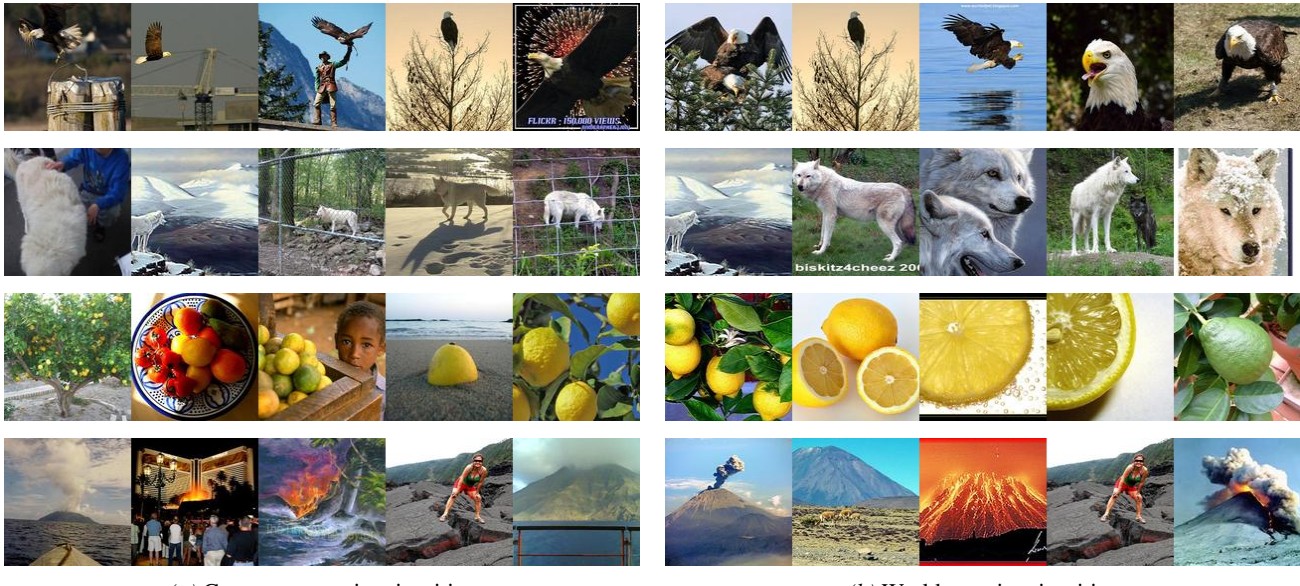

(a) Generator-centric minorities                    (b) World-centric minorities

*Figure 4.* **Comparison of minority samples under generator-centric (a) and world-centric (b) definitions on ImageNet** $256 \times 256$**.** We visualize minority samples across four ImageNet classes: bald eagle (top row), white wolf (second row), lemon (third row), and volcano (bottom row). Generator-centric minorities are defined by AvgkNN distance in the training set, while world-centric minorities are determined by JEPA-SCORE with DINOv2 (Oquab et al., 2023). Due to the large scale of the dataset, the two definitions exhibit partial overlap (Balestriero et al., 2025). However, a key qualitative difference emerges: generator-centric minorities tend to capture atypical *contexts* (*e.g.*, an eagle near construction equipment), whereas world-centric minorities more often exhibit atypical *behavior or appearance* (*e.g.*, mating eagles, unusually shaped lemons).

## B.2. Analysis of Approximation Error

In this section, we provide an empirical analysis of the JEPA-SCORE approximation. Following Proposition 4.1, we decompose the approximation error $\text{JS}(\boldsymbol{x}) - \bar{\text{JS}}(\boldsymbol{x})$ into two terms:

$$\text{JS}(\boldsymbol{x}) - \bar{\text{JS}}(\boldsymbol{x}) = \sum_{i=1}^{r} \log\big(\sigma_i\big(J_f(\boldsymbol{x})\big)\big) - \sum_{i=1}^{k} \log\big(\tilde{\sigma}_i\big(\bar{J}_f(\boldsymbol{x})\big)\big)$$

$$= \underbrace{\sum_{i=1}^{k} \log\left(\frac{\sigma_i\big(J_f(\boldsymbol{x})\big)}{\tilde{\sigma}_i\big(\bar{J}_{f,k}(\boldsymbol{x})\big)}\right)}_{=: \, e_{\text{RSVD}}(\boldsymbol{x})} + \underbrace{\sum_{i=k+1}^{r} \log\big(\sigma_i\big(J_f(\boldsymbol{x})\big)\big)}_{=: \, e_{\text{Trunc}}(\boldsymbol{x})},$$

where $e_{\text{RSVD}}(\boldsymbol{x})$ denotes the estimation error of top-$k$ singular values due to randomized SVD, and $e_{\text{Trunc}}(\boldsymbol{x})$ is the truncation error from discarding $\{\sigma_i\}_{i=k+1}^{r}$. As discussed in Section 4.2, the estimation error $e_{\text{RSVD}}(\boldsymbol{x})$ can be made small with a few power iterations, *e.g.*, $q = 2$ (see Table 5a). While the truncation error $e_{\text{Trunc}}(\boldsymbol{x})$ can be large for small $k$, we empirically find that it is dominated by an image-agnostic *offset* that contributes little to distinguishing atypicality across samples. To illustrate this, we further decompose the truncation error as:

$$e_{\text{Trunc}}(\boldsymbol{x}) = e_{\text{Trunc}}^{\Delta}(\boldsymbol{x}) + \Delta_{\text{Offset}}, \tag{24}$$

where $e_{\text{Trunc}}^{\Delta}(\boldsymbol{x})$ denotes the semantic component that captures image-dependent information, and $\Delta_{\text{Offset}}$ is an image-agnostic offset term.

**Singular Value Spectrum.** Figure 5a shows the singular-value spectrum of the DINOv2 Jacobian on 1k CelebA-64 real samples. Note that the variance of leading singular values is strongly sample-dependent, while the tail becomes nearly constant across images (Figures 5b and 5c). This is further supported by Figure 5d: the cumulative variance ratio reaches 78.7% at $k = 9$, indicating that the top-$k$ components capture most of the image-dependent variation, while the remaining components contribute little. The variance exhibits a clear elbow around $k = 9$, beyond which additional components contribute negligible variance, suggesting that components beyond a threshold $k_{\text{th}}$ are dominated by noise-like

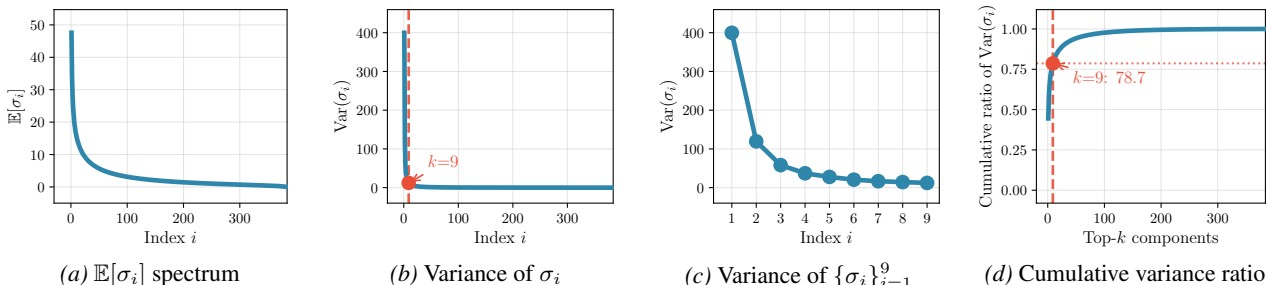

*Figure 5.* **Singular value spectrum on CelebA-64 using DINOv2** (Oquab et al., 2023). **(a)** $\mathbb{E}[\sigma_i]$ decays sharply for small $i$ and forms a long tail. **(b)** $\mathrm{Var}(\sigma_i)$ drops rapidly and plateaus after the elbow (red marker at $k = 9$), indicating that leading components capture image-dependent characteristics while the tail behaves as a near-constant offset. **(c)** Enlarged view of **(b)** for $i \leq 9$. **(d)** Cumulative variance ratio reaches 78.7% at $k = 9$, confirming that the top-9 components account for most of the variance.

| $k$ | $\mathrm{JS} - \bar{\mathrm{JS}}$ | $e_{\mathrm{RSVD}}$ | $e_{\mathrm{Trunc}}$ | $e^{\Delta}_{\mathrm{Trunc}}$ | $\mathcal{E}_{\mathrm{RSVD}}$ | $\mathcal{E}_{\mathrm{Trunc}}$ | $R^{\Delta}_{\mathrm{trunc}}$ |
|---|---|---|---|---|---|---|---|
| 2 | 159.92 | 0.020 | 159.90 | 31.64 | 2.55 | 581.78 | 80.21% |
| 4 | 153.51 | 0.053 | 153.46 | 25.20 | 4.83 | 547.28 | 83.58% |
| 6 | 147.52 | 0.086 | 147.43 | 19.16 | 7.05 | 520.82 | 87.00% |
| 8 | 141.81 | 0.123 | 141.69 | 13.43 | 9.21 | 498.50 | 90.53% |
| 10 | 136.33 | 0.157 | 136.18 | 7.91 | 11.31 | 478.76 | 94.19% |

*(a)* Real samples $\boldsymbol{x}_0$

| $k$ | $\mathrm{JS} - \bar{\mathrm{JS}}$ | $e_{\mathrm{RSVD}}$ | $e_{\mathrm{Trunc}}$ | $e^{\Delta}_{\mathrm{Trunc}}$ | $\mathcal{E}_{\mathrm{RSVD}}$ | $\mathcal{E}_{\mathrm{Trunc}}$ | $R^{\Delta}_{\mathrm{trunc}}$ |
|---|---|---|---|---|---|---|---|
| 2 | 38.22 | 0.032 | 38.18 | 27.68 | 2.67 | 441.22 | 27.51% |
| 4 | 32.64 | 0.071 | 32.57 | 22.06 | 5.06 | 412.51 | 32.26% |
| 6 | 27.39 | 0.104 | 27.29 | 16.78 | 7.35 | 388.91 | 38.50% |
| 8 | 22.39 | 0.144 | 22.26 | 11.75 | 9.56 | 368.59 | 47.21% |
| 10 | 17.60 | 0.180 | 17.43 | 6.92 | 11.71 | 350.31 | 60.29% |

*(b)* Denoised predictions $\hat{\boldsymbol{x}}_{0|t}$

*Table 5.* **Approximation error decomposition.** **(a)** Measured on 1k real CelebA-64 samples. **(b)** Measured on denoised predictions $\hat{\boldsymbol{x}}_{0|t}$ during inference. $R^{\Delta}_{\mathrm{trunc}}$ denotes the offset ratio $\Delta_{\mathrm{Offset}}/e_{\mathrm{Trunc}} \times 100\%$.

variations (*i.e.*, offset) rather than semantic characteristics. On SDXL-Lightning, we observe diminishing returns for $k > 13$ (see Table 3c), motivating us to use $k_{\mathrm{th}} = 13$ for the following analysis. Accordingly, we define the offset term in Eq. (24) as the log-sum of tail singular values:

$$\Delta_{\mathrm{Offset}} \coloneqq \sum_{i=k_{\mathrm{th}}}^{r} \log\left(\sigma_i(J_f(\boldsymbol{x}))\right).$$

**Error Analysis.** Table 5a shows the approximation error $\mathrm{JS}(\boldsymbol{x}) - \bar{\mathrm{JS}}(\boldsymbol{x})$ measured on 1k CelebA-64 real samples. Several trends emerge as $k$ increases: (i) the total error decreases monotonically, confirming that higher-rank approximations better recover JEPA-SCORE; (ii) $e_{\mathrm{RSVD}}$ remains negligible, indicating accurate singular value estimation; (iii) $e_{\mathrm{Trunc}}$ dominates and decreases with $k$; (iv) the theoretical bounds $\mathcal{E}_{\mathrm{RSVD}}$ and $\mathcal{E}_{\mathrm{Trunc}}$ correctly upper-bound their empirical counterparts. Importantly, $e^{\Delta}_{\mathrm{Trunc}}$ decreases with $k$, indicating that larger $k$ preserves more semantic information. Since $\Delta_{\mathrm{Offset}}$ is constant by definition, the offset ratio $R^{\Delta}_{\mathrm{trunc}}$ increases with $k$. This confirms that as $k$ grows, the truncation error becomes increasingly dominated by the image-agnostic offset, while the semantic component diminishes. We see the same trend of the approximation error measured during inference time (on $\hat{\boldsymbol{x}}_{0|t}$); see Table 5b for details.

### B.3. Computational Analysis

Table 6 compares the computational cost of JEPA guidance against baselines on SDv1.5. While our method requires additional overhead from Jacobian computation and randomized SVD, it remains faster than MinorityPrompt (Um & Ye, 2025) while achieving substantially lower JEPA-SCORE with comparable quality and text-alignment metrics. Compared to SGMS (Um & Ye, 2024), our method incurs higher cost due to the JEPA encoder forward pass and SVD operations at each guidance step,

| Method | Time ↓ | CLIP ↑ | Pick ↑ | ImgR ↑ | JEPA ↓ |
|---|---|---|---|---|---|
| DDIM | 1.45 | 31.52 | 21.49 | 0.21 | -292.27 |
| CADS | **1.44** | 31.46 | 21.28 | 0.09 | -311.23 |
| SGMS | 7.24 | 31.23 | 21.32 | 0.12 | -311.14 |
| MinorityPrompt | 12.42 | **31.56** | 21.32 | **0.24** | -322.33 |
| Ours | 10.44 | 31.46 | **21.50** | 0.22 | **-355.40** |

*Table 6.* **Computational cost comparison.** Time denotes seconds per sample. Best in **bold**, second best underlined.

which could be justified by the significant improvement in world-centric atypicality. We note that samplers like DDIM and CADS are faster as they do not involve any guidance computation. Overall, JEPA guidance offers a reasonable trade-off between computational cost and the ability to generate world-centric minority samples.

## C. Implementation Details

**Baselines.** For all baselines, we follow the official implementations (if available) or descriptions provided in existing papers. For Sehwag et al. (2022), we use the same pretrained models as ours (*i.e.*, ADM checkpoints (Dhariwal & Nichol, 2021)). Specifically on CelebA, following previous works (Um et al., 2024; Um & Ye, 2024; Um et al., 2025), we train an out-of-distribution (OOD) classifier to distinguish whether an input belongs to CelebA or other datasets (*e.g.*, ImageNet), and incorporate the negative log-likelihood gradient targeting the in-distribution class into DDPM sampling Eq. (3) for low-density guidance. We implement MG (Um et al., 2024) with the official codebase[2] by employing U-Net encoders for minority classifiers trained on the entire training set. For ImageNet $256 \times 256$, we use the upscaling model (Dhariwal & Nichol, 2021) as described in Um et al. (2024). We implement SGMS (Um & Ye, 2024) and BnS (Um et al., 2025) by following their official implementations[3]. For CADS (Sadat et al., 2023), we implement based on the pseudocode in the paper. MinorityPrompt (Um & Ye, 2025) is implemented using the official codebase[4].

**Evaluation Metrics.** Our evaluation protocol follows previous works (Um et al., 2024; Um & Ye, 2024; Um et al., 2025). Specifically, we evaluate clean Fréchet Inception Distance (cFID) (Parmar et al., 2022) and Spatial FID (sFID) (Nash et al., 2021) using official implementations[5]. For sFID, we use spatial features (*i.e.*, the first 7 channels from `mixed_6/conv`) instead of the standard `pool_3` inception features. For Improved Precision & Recall (Kynkäänniemi et al., 2019), we follow the implementation in Han et al. (2022) with $k = 5$. Density & Coverage (Naeem et al., 2020) are computed using the official implementation[6]. To evaluate closeness to world-centric minorities, we use samples with the lowest JEPA-SCORE—the most atypical under the world prior—as reference data. Specifically, we select the 10K real samples with the lowest JEPA-SCORE from CelebA and ImageNet training sets. CLIPScore is computed using `torchmetrics`[7]. For PickScore and ImageReward, we use the official implementations[8]. For JEPA-SCORE, we use the JEPA encoder of `DINOv2-ViT-S/14` (Oquab et al., 2023). All metrics are computed on 30K generated samples for unconditional and class-conditional generation, and 5K samples for text-conditional generation.

**Hyperparameters.** We use 250 sampling steps for CelebA and ImageNet, and 50 steps for SDv1.5 and 4 steps for SDXL-Lightning. For randomized SVD, we set $(k, p, q) = (3, 2, 2)$ for CelebA and ImageNet, and $(k, p, q) = (9, 2, 2)$ for SDv1.5 and SDXL-Lightning. The deferred guidance ratio is set to $\tau = 0.8$ across all settings. The intermittent rate is $N = 3$, except for SDXL-Lightning where $N = 1$. We use the variance schedule $\eta_t = \eta \Sigma_\theta(x_t, t)$ for CelebA and ImageNet, and the constant schedule $\eta_t = \eta$ for SDv1.5 and SDXL-Lightning. Guidance strength is $\eta = 2.0$ for CelebA and ImageNet, $\eta = 0.06$ for SDv1.5, and $\eta = 0.5$ for SDXL-Lightning. The JEPA encoder $J_f$ is `DINOv2-ViT-S/14` (Oquab et al., 2023) for all experiments. Our implementation is based on PyTorch (Paszke et al., 2019). All experiments were conducted on NVIDIA A100 GPUs. Code is available at `https://github.com/soobin-um/jepa-guidance`.

## D. Additional Experimental Results

### D.1. User Study Results

To complement our quantitative evaluation on text-conditional generation, we conducted a human preference study comparing JEPA guidance against three baselines: (i) DDIM (Song et al., 2020a), (ii) SGMS (Um & Ye, 2024), and (iii) MinorityPrompt (Um & Ye, 2025). A total of 31 participants evaluated randomly ordered image pairs generated with identical prompts and seeds. For each pair, participants selected the preferred sample based on two criteria: (i) *Alignment & Quality* (text adherence

|                      | Alignment & Quality | | Atypicality | |
|----------------------|:--------:|:------:|:--------:|:------:|
|                      | Baseline | Ours   | Baseline | Ours   |
| vs. DDIM             | **52.90** | 47.10  | 18.71    | **81.29** |
| vs. SGMS             | 34.19    | **65.81** | 21.29  | **78.71** |
| vs. MinorityPrompt   | 33.55    | **66.45** | 22.58  | **77.42** |

*Table 7.* **Human preference evaluation (%).** Percentage of user preference for each method.

and visual fidelity) and (ii) *Atypicality* (semantic distinctiveness). As shown in Table 7, JEPA guidance is consistently preferred over both baselines across both criteria, demonstrating its effectiveness in generating high-quality atypical samples.

---

[2]`https://github.com/soobin-um/minority-guidance`
[3]`https://github.com/soobin-um/sg-minority`, `https://github.com/soobin-um/BnS`
[4]`https://github.com/soobin-um/MinorityPrompt`
[5]`https://github.com/GaParmar/clean-fid`, `https://github.com/mseitzer/pytorch-fid`
[6]`https://github.com/clovaai/generative-evaluation-prdc`
[7]`https://lightning.ai/docs/torchmetrics/stable/multimodal/clip_score.html`
[8]`https://github.com/yuvalkirstain/PickScore`, `https://github.com/THUDM/ImageReward`

## D.2. Additional Generated Samples

We present additional generated samples across all datasets for qualitative comparison; see Figures 6 to 9. Samples generated by our approach capture world-level atypicality compared to previous minority sampling methods based on generative priors. We also observe that MinorityPrompt (Um & Ye, 2025) sometimes exhibits quality degradation on SDv1.5, such as noisy artifacts (see the 3rd row, 2nd column in Figure 8). CLIPScore and ImageReward often fail to capture such deterioration, while PickScore does, resulting in low PickScore values for MinorityPrompt (Table 2). In contrast, our JEPA guidance does not exhibit such degradation, producing high-fidelity images with world-level atypical features.

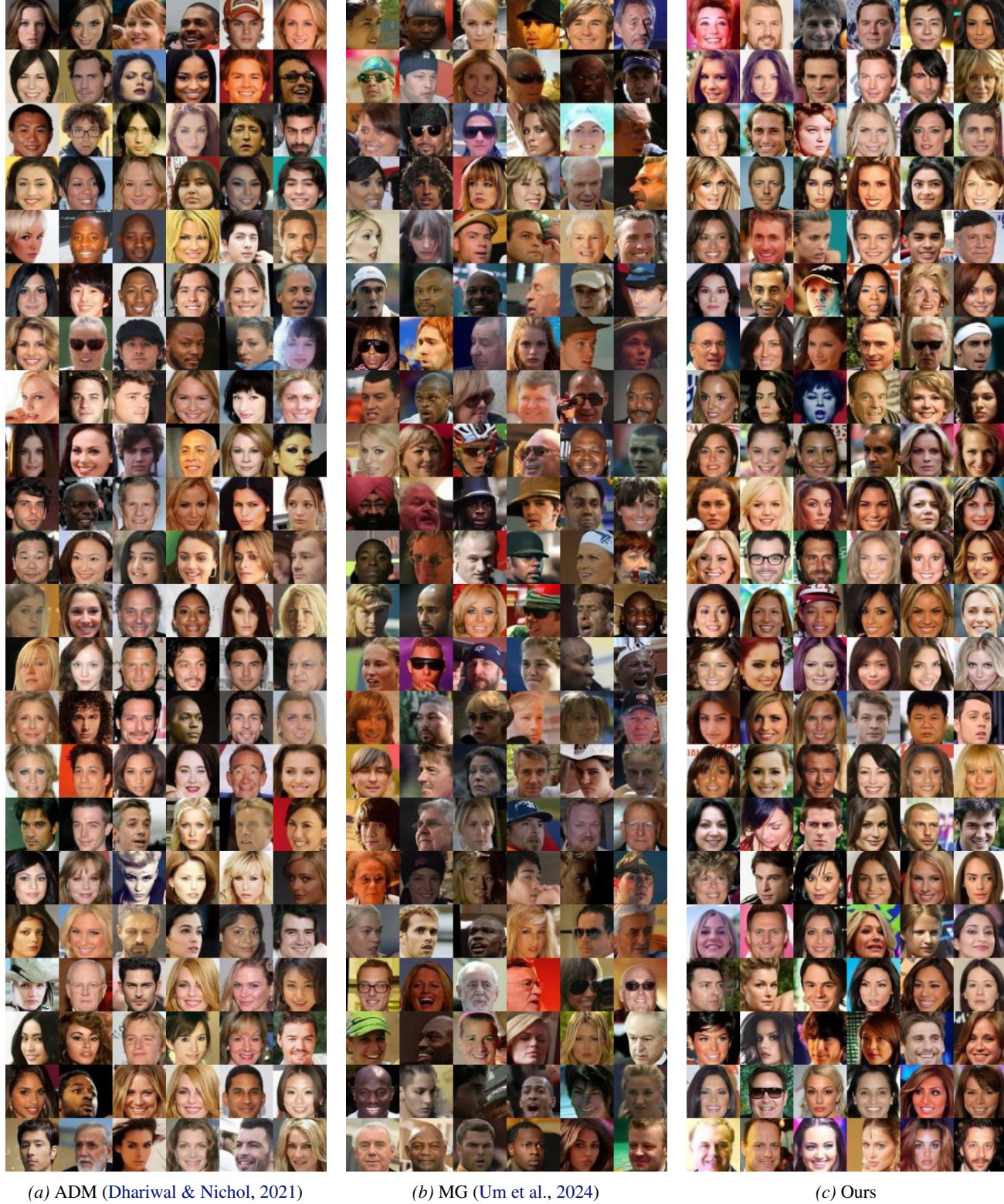

*(a)* ADM (Dhariwal & Nichol, 2021)  *(b)* MG (Um et al., 2024)  *(c)* Ours

*Figure 6.* **Sample comparison on CelebA** $64 \times 64$**.**

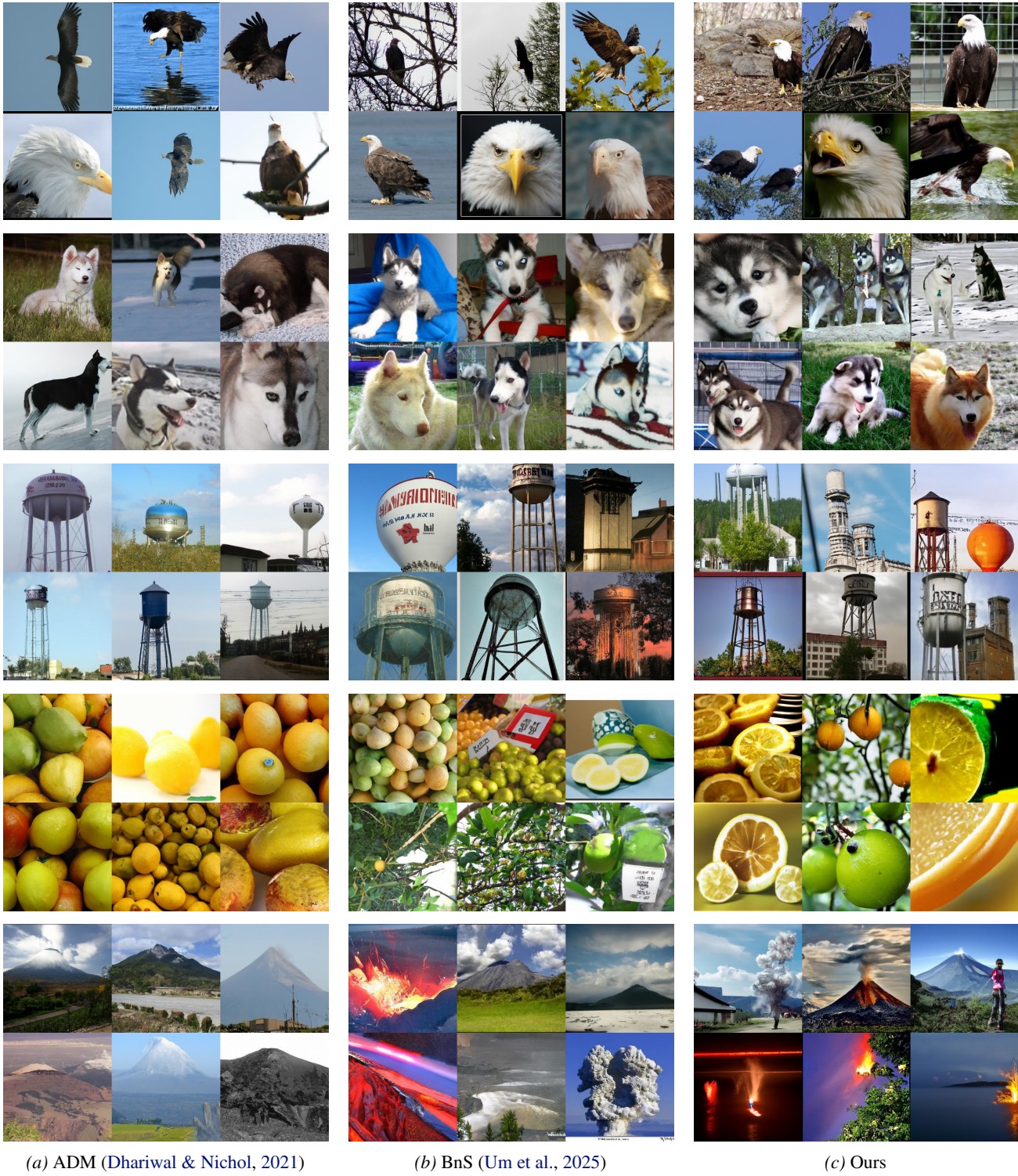

*(a)* ADM (Dhariwal & Nichol, 2021)  *(b)* BnS (Um et al., 2025)  *(c)* Ours

*Figure 7.* **Sample comparison on ImageNet** $256 \times 256$**.** Generated samples from four classes: (i) "bald eagle" (top row); (ii) "Siberian husky" (second row); (iii) "water tower" (third row); (iv) "lemon" (bottom row).

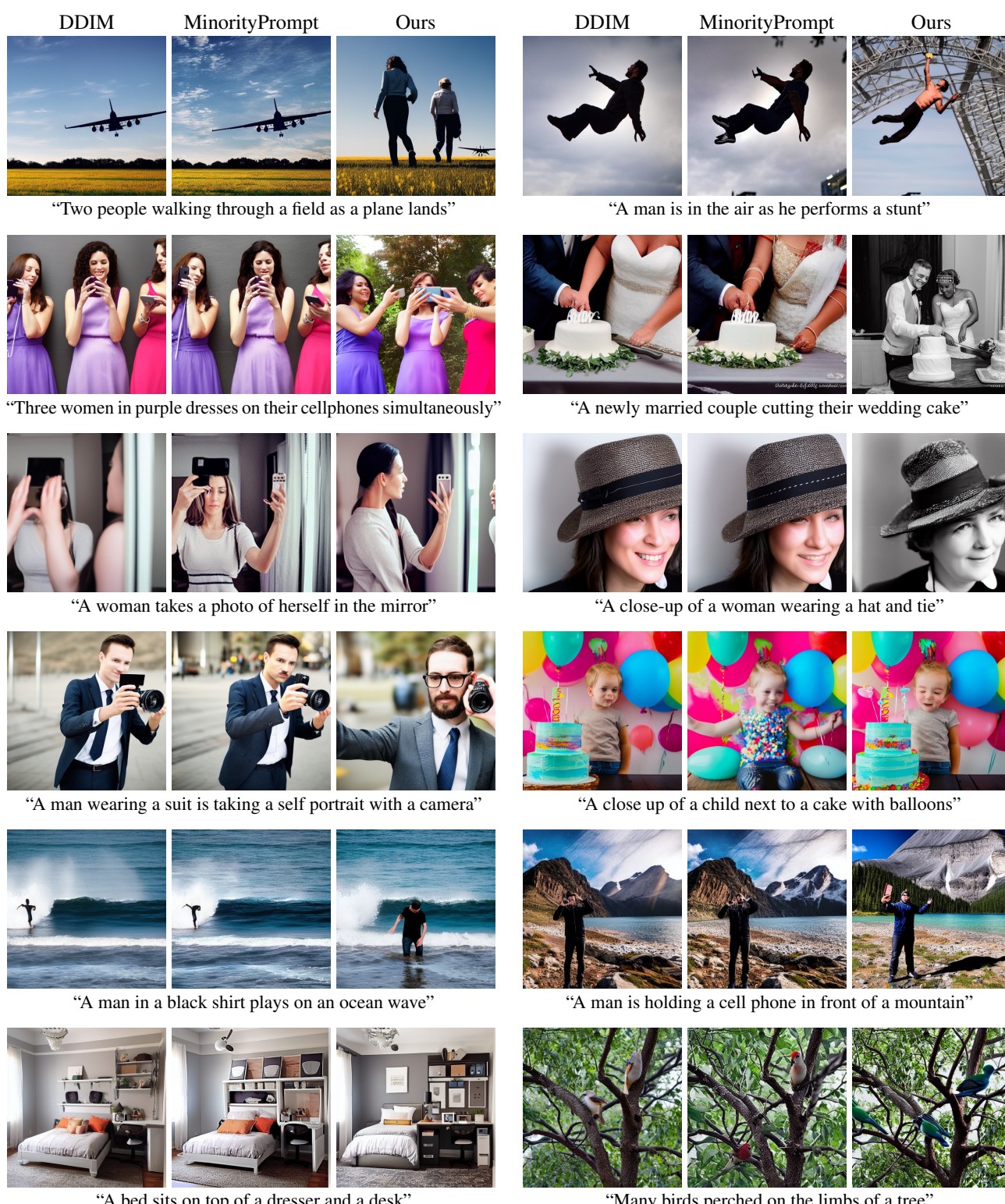

DDIM     MinorityPrompt     Ours        DDIM     MinorityPrompt     Ours

"Two people walking through a field as a plane lands"      "A man is in the air as he performs a stunt"

"Three women in purple dresses on their cellphones simultaneously"      "A newly married couple cutting their wedding cake"

"A woman takes a photo of herself in the mirror"      "A close-up of a woman wearing a hat and tie"

"A man wearing a suit is taking a self portrait with a camera"      "A close up of a child next to a cake with balloons"

"A man in a black shirt plays on an ocean wave"      "A man is holding a cell phone in front of a mountain"

"A bed sits on top of a dresser and a desk"      "Many birds perched on the limbs of a tree"

*Figure 8.* **Sample comparison on SDv1.5.** Generated samples from three approaches: (i) DDIM (Song et al., 2020a), (ii) MinorityPrompt (Um & Ye, 2025), and (iii) Ours. Six prompts were used, and random seeds were shared across all methods.

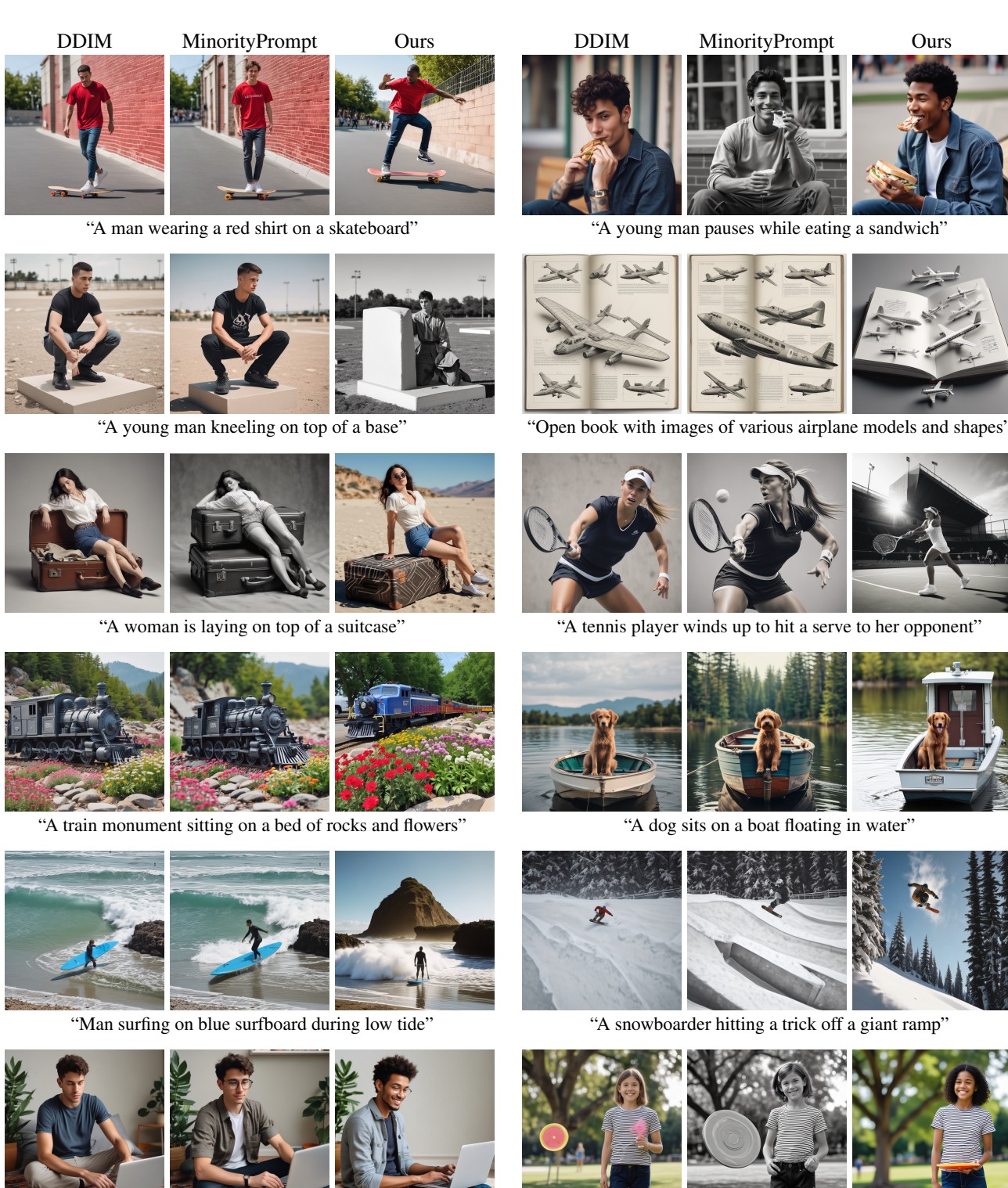

DDIM     MinorityPrompt     Ours        DDIM     MinorityPrompt     Ours

"A man wearing a red shirt on a skateboard"      "A young man pauses while eating a sandwich"

"A young man kneeling on top of a base"      "Open book with images of various airplane models and shapes"

"A woman is laying on top of a suitcase"      "A tennis player winds up to hit a serve to her opponent"

"A train monument sitting on a bed of rocks and flowers"      "A dog sits on a boat floating in water"

"Man surfing on blue surfboard during low tide"      "A snowboarder hitting a trick off a giant ramp"

"A young man is sitting on a rug while on a laptop"      "A young girl holds a Frisbee at a park"

*Figure 9.* **Additional sample comparison on SDXL-Lightning.** Generated samples from three approaches: (i) DDIM (Song et al., 2020a), (ii) MinorityPrompt (Um & Ye, 2025), and (iii) Ours. Six prompts were used, and random seeds were shared across all methods.

