# OpenReview forum: "Beyond Generative Priors: Minority Sampling with JEPA-Guided Diffusion"
_ICML.cc/2026/Conference — ICML 2026 regular_

### Official Review · Reviewer_WVET · 2026-03-04

**Soundness:** 2
**Presentation:** 3
**Significance:** 3
**Originality:** 3
**Overall Recommendation:** 4
**Confidence:** 4

**Summary:**

This paper redefines minority sampling from "rare under the generative model's prior" to "rare under a world model's prior," using JEPA-SCORE. They guide diffusion sampling toward low JEPA-SCORE regions using randomized SVD approximations for faster inference.

**Compliance With Llm Reviewing Policy:**

Affirmed.

**Final Justification:**

After considering the authors' reply, I am happy to raise my initial scoring.

**Key Questions For Authors:**

Please address the weaknesses above. In addition, could the authors provide answers to the following questions?

1) How does randomized SVD make guidance faster exactly? You still need to compute the Jacobian?
2) What is the practical takeaway of Proposition 4.1? The bound decomposes error into RSVD error and truncation error but what is its relevance in the paper?

**Limitations:**

yes

**Strengths And Weaknesses:**

**Strengths:**
- Decoupling the definition of minority from the generative model and grounding it in an external representation is a well-motivated idea.
- The method works at inference time with off-the-shelf models, requiring no retraining.

**Weaknesses:**
- Calling JEPA a "world model" is a stretch and in my opinion overselling. The "world-centric" minorities are really DINOv2-training-data-centric minorities.
- JEPA-SCORE is used both as the guidance objective and as the primary evaluation metric. Of course the method achieves the lowest JEPA-SCORE (Table 1 + 2).
- Besides JEPA-Score, which is questionable with the aforementioned point, the overall quantiative results are modest and within a narrow range.

---

> ### Author Rebuttal · Authors · 2026-03-30
>
> We sincerely thank Reviewer `WVET` for the insightful comments and feedback. Below we provide point-by-point responses addressing your concerns.
>
> ---
>
> > **W1. `WVET` raised concerns that calling JEPA a "world model" is overselling, and that the proposed world-centric minorities are essentially DINOv2-training-data-centric.**
>
> We appreciate this perspective. Our use of "world model" follows the terminology of LeCun (2022) [1], who positions JEPAs as a promising class of world models. We acknowledge that the term may overstate the current capabilities of JEPAs, and we are happy to soften the language in the revised manuscript (e.g., "JEPA-centric" or "representation-centric").
>
> That said, we note that the JEPA encoder mainly used in our work (DINOv2) is trained on LVD-142M, a large-scale dataset spanning 142M diverse images, capturing substantially broader visual semantics than typical generator training sets (e.g., CIFAR-10 with 50K images); see Figures 2 and 4 for visualization. In this sense, the JEPA prior serves as a meaningfully broader proxy for real-world data density compared to generator-centric priors, which is the core distinction our work aims to highlight.
>
> ---
>
> > **W2. `WVET` noted that using JEPA-SCORE as both the guidance objective and the evaluation metric makes the results inherently favorable to our approach.**
>
> Please refer to our response to Reviewer `zDno` (W3-1).
>
> ---
>
> > **W3. `WVET` noted that the overall quantitative results are modest and within a narrow range.**
>
> We would like to assure the reviewer that the quantitative improvements are substantial. In Table 2, our approach outperforms all minority/diversity approaches except MinorityPrompt in quality metrics on SDv1.5, while achieving substantially lower JEPA-SCORE. Regarding MinorityPrompt, we note that it often produces noisy artifacts in generated samples (see Figure 8 and Section D.2), which is reflected in its notably lower PickScore. On SDXL-Lightning, our approach achieves the best or comparable quality metrics across the board while maintaining the lowest JEPA-SCORE. While SGMS yields low JEPA-SCORE competitive to ours, it sacrifices significant alignment/quality, as reflected in its considerably lower CLIPScore, PickScore, and ImageReward.
>
> We note that some compromise relative to standard samplers like DDIM is expected, as minority generation inherently involves a trade-off between quality and low-density generation [2-4]. While PRDC metrics occasionally fall below some baselines, these are secondary in T2I generation due to the inherent domain gap between MS-COCO reference data and generated samples; see our response to Reviewer `zDno` for details on this point (W3-2). The notable improvements are further supported by our human preference study (Table 7), where users prefer our samples in both quality and atypicality.
>
> ---
>
> > **Q1. `WVET` questioned how randomized SVD makes guidance faster when the Jacobian still needs to be computed.**
>
> Good question. The Jacobian computation is indeed required in ours as well (Line 8 in Algorithm 1). The speedup occurs in the subsequent steps. Unlike exact JEPA-SCORE guidance, which requires SVD on the full Jacobian $J\_f$ (notoriously expensive for large-scale models), our approach first obtains a compact projection matrix $Q$ (L9 in Alg. 1) and performs SVD on the much smaller compressed matrix $Q^{\top} J\_{f}$. Furthermore, the envelope theorem (Section 4.2) allows us to treat $Q$ as a constant during backpropagation (L10 in Alg. 1), eliminating the need to differentiate through the SVD procedure. Together, these reduce both the forward and backward computational cost substantially. See the runtime comparison in our response to Reviewer `zDno` (W4) for quantitative details.
>
> ---
>
> > **Q2. `WVET` questioned the practical relevance of Proposition 4.1.**
>
> Proposition 4.1 provides practical guidelines for selecting key hyperparameters. The RSVD error term indicates that with appropriate oversampling $p$ and power iterations $q$ (e.g., $p=2$, $q=2$), the top-$k$ singular value approximation via randomized SVD is accurate; see Table 5 for demonstration. The truncation error term suggests that rank $k$ should not be too small, as discarded singular values contribute to the approximation gap. Fortunately, our analysis in Section B.2 reveals that these discarded components are largely dominated by an image-agnostic offset that contributes little to distinguishing atypicality across samples, meaning that $k \approx 9-10$ often suffices in practice. This is also empirically validated in Table 3c, where we observe diminishing returns beyond $k=9$.
>
> ---
>
> **References**
>
> [1] A Path Towards Autonomous Machine Intelligence, OpenReview, 2022.
>
> [2] Don't Play Favorites: Minority Guidance for Diffusion Models, ICLR 2024.
>
> [3] Minority-Focused Text-to-Image Generation via Prompt Optimization, CVPR 2025.
>
> [4] Boost-and-Skip: A Simple Guidance-Free Diffusion for Minority Generation, ICML 2025.

---

> > ### Author Rebuttal · Reviewer_WVET · 2026-04-05
> >
> > Thank you for your responses, I am willing to upscore my rating.

---

> > > ### Author Response · Authors · 2026-04-05
> > >
> > > Thank you for acknowledging our rebuttal and for considering an increase in your score. We are glad that our response fully addressed your concerns. Your constructive feedback is greatly appreciated.

---

### Official Review · Reviewer_FUa4 · 2026-03-12

**Soundness:** 3
**Presentation:** 3
**Significance:** 3
**Originality:** 2
**Overall Recommendation:** 5
**Confidence:** 3

**Summary:**

The paper introduces a new method on minority sampling by redefining minorities with respect to world priors rather than generator-centric priors. The authors also tried approximation method to reduce the computation complexity for computation of the JEPA prior. The proposed JEPA guidance empirically shows that the method is somehow able to generate some more diverse low-likelihood samples during the samples.

**Compliance With Llm Reviewing Policy:**

Affirmed.

**Final Justification:**

The rebuttal has fulled addressed my concerns and I increase score from weak accept to accept.

**Key Questions For Authors:**

Please see the above weakness for questions. Please clarify during the rebuttal and correct me if I misunderstood any thing.

**Limitations:**

Yes, The authors did.

**Strengths And Weaknesses:**

Strength:

1.	The paper proposes to use the JEPA score as the target metric in order to get minority samples from during the generations.
Instead of directly using JEPA-SCORE at each sampling step and use its gradient as a guidance signal, the paper proposes to use the technique randomized SVD to approximate JEPA-SCORE.
2.	The paper overcomes many computation complexity issues such as memory issues via existing method such as Envelope Theorem. By relying on these method, the paper successfully implement the new method by adjusting an d approximating their original idea.

3.	Empirical result somehow demonstrates the new method is effective in achieving the boosted performance.


Weakness:

1.	My biggest concern is that the empirical study shows the generated sample somehow is deviating from the intended distribution. Especially when you look at Figure 1, the prompt : “Three persons with military attire seated on a bench”, the generated figure by ht proposed method is not adhering to only generating 3 people. Rather, it generates 5 people. Although strictly speaking, this could be owing to the method treating the 2 extra people as background, still, this leaves the concern that the proposed method is steering the sampling distribution away from the intended distribution too much.

2.	It seems the proposed method is just using existing methods and combine them to achieve the goal of minority sampling. I do not see why JEPA prior is necessary to achieve the goal of minority sampling. Please clarify in principle, why particularly JEPA prior should be preferred over vanilla diffusion model for the purpose of minority sampling, and why not just sample from low likelihood region from the learned distribution out of diffusion model? I do not see the theoretical support in this part or any related discussions. It could be interesting to add this discussion. Also, does the method generalize to any other SSL encoders (JEPA is one special case of them with the goal to learn useful embeddings, I believe)?

3.	In Table 4. the accuracy of the classifiers trained using the augmented generated by the proposed method is not beating the baselines. This is surprising as the method seems to provide larger diversity and variance of the training data, could the authors analyze why?

---

> ### Author Rebuttal · Authors · 2026-03-30
>
> We sincerely thank Reviewer `FUa4` for the insightful comments and constructive feedback. Below we provide point-by-point responses addressing your concerns.
>
> ---
>
> > **W1. `FUa4` raised concerns that the proposed approach potentially yields deviations from the intended distribution.**
>
> We respectfully note that the prompt specifies three persons seated on a bench, and our sample indeed depicts exactly three seated individuals; the two standing figures in the background are not constrained by the prompt. We also note that MinorityPrompt similarly generates an additional standing figure in the background.
>
> More importantly, our text-alignment metrics (CLIPScore, PickScore, ImageReward) remain comparable to or higher than baselines (Table 2), quantitatively confirming that JEPA guidance does not systematically deviate from the intended distribution. This is further supported by our human preference study (Table 7), where users prefer our samples in text-alignment and quality (as well as in atypicality).
>
> ---
>
> > **W2-1. `FUa4` raised concerns about the novelty of the proposed approach and questioned why JEPA prior is necessary for minority sampling.**
>
> The reviewer is kindly reminded that our key contributions lie in: (i) establishing a new perspective on minority sampling with respect to world-model priors, and (ii) providing practical techniques to realize this perspective.
>
> As discussed in Section 4.1 and Figure 2 (extended in Section B.1), existing minority samplers [1-4] operate solely under generative priors, yielding low-density samples with respect to learned data distributions. The problem is that when the training data is limited in scale (which is often the case in reality), minority samples drawn from the generative distribution do not necessarily correspond to genuinely rare instances in the real world (see Figure 2). This is precisely why we introduce JEPA prior: JEPAs have been recognized as a promising class of world models (LeCun, 2022) [5], and their representations implicitly encode real-world data density through JEPA-SCORE [6], enabling us to explicitly define and target world-centric minorities that are semantically more meaningful (see Figures 2 and 4). The practical benefit is further evidenced by our downstream application (Table 4), where augmentation with our samples achieves the best F1, Precision, and Recall using only 30K samples, compared to 50K for all baselines.
>
> Regarding the second point, while the individual techniques (e.g., randomized SVD, envelope theorem) are known, incorporating them to make JEPA-SCORE guidance computationally practical, together with the theoretical guarantee in Proposition 4.1 and the empirical analysis in Section B.2, is non-trivial.
>
> ---
>
> > **W2-2. `FUa4` questioned whether the proposed approach generalizes to other SSL encoders beyond JEPA.**
>
> Thanks for pointing this out. We have demonstrated in Table 3d (in our manuscript) that our guidance framework also operates with MetaCLIP, a multimodal model outside the typical JEPA family, suggesting broader applicability beyond JEPA encoders. We further show in our response to Reviewer `PUvi` that DINOv3 can also serve as the guiding encoder; see details therein.
>
> ---
>
> > **W3. `FUa4` raised concerns that the classification accuracy from augmentation with our generated samples does not outperform all baselines in Table 4.**
>
> Thank you for this insightful question. We would like to clarify that our method achieves highly competitive accuracy (0.902 vs. 0.903 for SGMS, a negligible difference of 0.001) while substantially outperforming all baselines in F1 (+0.018) and achieving the best Precision and Recall. We argue that F1, Precision, and Recall are more appropriate metrics for evaluating augmentation quality in this setting, as accuracy tends to be dominated by majority classes and can obscure improvements in underrepresented attributes. The consistent gains across these balanced metrics suggest that the diversity of our generated samples translates into more informative training data, contributing to balanced and generalizable decision boundaries. We also note that our method achieves these results using only 30K augmented samples, compared to 50K for all baselines (40% reduction), demonstrating superior data efficiency.
>
> ---
>
> **References**
>
> [1] Don't Play Favorites: Minority Guidance for Diffusion Models, ICLR 2024.
>
> [2] Self-Guided Generation of Minority Samples Using Diffusion Models, ECCV 2024.
>
> [3] Minority-Focused Text-to-Image Generation via Prompt Optimization, CVPR 2025.
>
> [4] Boost-and-Skip: A Simple Guidance-Free Diffusion for Minority Generation, ICML 2025.
>
> [5] A Path Towards Autonomous Machine Intelligence, OpenReview, 2022.
>
> [6] Gaussian Embeddings: How JEPAs Secretly Learn Your Data Density, arXiv 2025.

---

> > ### Author Rebuttal · Reviewer_FUa4 · 2026-04-01
> >
> > Thanks for the clarification. I will therefore raise my score from weak accept to accept.

---

> > > ### Author Response · Authors · 2026-04-01
> > >
> > > Thank you for acknowledging our rebuttal and for raising the score. We are happy that our response fully addressed your previous concerns. We appreciate your constructive feedback, which helped strengthen our work.

---

### Official Review · Reviewer_PUvi · 2026-03-12

**Soundness:** 3
**Presentation:** 4
**Significance:** 4
**Originality:** 4
**Overall Recommendation:** 6
**Confidence:** 4

**Summary:**

This paper proposes a JEPA (world-model) based approach for minority sampling in diffusion models. Minority sampling aims to generate samples from low-density regions of the data distribution. Existing approaches are constrained by the implicit prior defined by the generative model itself, and therefore cannot generate samples that are semantically rare in the real world. To address this limitation, the authors propose a minority sampling method guided by JEPA, a model that captures semantic structures of the real world. The proposed method uses a guidance signal derived from a metric called the JEPA-score, which estimates the real-world data density. Furthermore, since computing the standard JEPA-score is computationally expensive, the authors introduce a practical approximation that enables efficient computation.

**Compliance With Llm Reviewing Policy:**

Affirmed.

**Final Justification:**

After carefully reviewing the paper, the authors' rebuttal, and the other reviews, I maintain my strong positive assessment and recommend acceptance.

The originality and significance of this work are outstanding. Most existing approaches to minority sampling rely on the implicit distribution of the generative model, which often fails to accurately reflect real-world semantic rarity. This paper elegantly addresses this limitation by introducing JEPA as a world model for minority guidance. Furthermore, the guidance design is simple and highly flexible, making it readily applicable to various settings, such as class-conditional and text-to-image generation. Crucially, as JEPA-style world models continue to advance, the effectiveness of this proposed approach will naturally scale with them, highlighting its broad potential impact.

The soundness of the methodology is also a major strength. The authors successfully overcome the computational bottleneck of JEPA-based guidance—computing the singular values of the Jacobian—by introducing a highly practical approximation using Randomized SVD. They empirically demonstrate that approximation errors from rank truncation do not negatively affect practical performance. This is backed by a comprehensive experimental evaluation across multiple datasets, tasks, and baselines. The thorough ablation studies and the demonstration of the generated samples' utility in downstream data augmentation further solidify the paper's claims.

Since I did not have any major concerns during the initial review phase, my primary focus during the rebuttal was observing the discussions. The authors' responses to other reviewers were convincing and only reinforced my confidence in the paper's quality.

Given its innovative use of world-centric sampling, practical efficiency, and rigorous empirical validation, I believe this is a highly impactful paper that fully meets the high standards of ICML.

**Key Questions For Authors:**

I do not have additional questions for the authors. The paper is generally clear and well explained.

**Limitations:**

yes

**Strengths And Weaknesses:**

Strengths
- First use of JEPA (a world model) for minority guidance: Most existing approaches to minority sampling rely on the distribution implicitly defined by the generative model itself. As a result, the generated samples do not necessarily reflect semantic rarity in the real world. The proposed method introduces JEPA as a world model for minority guidance, enabling sampling that better aligns with real-world semantic rarity.
- Simple and flexible guidance design: The proposed method introduces minimal restrictions beyond the use of a JEPA model. As a result, it can be applied to various generative settings, such as class-conditional generation and text-to-image generation. Furthermore, as JEPA-style world models continue to improve, the effectiveness of the proposed approach is likely to increase as well.
- Practical and efficient approximation of the JEPA score: A key limitation of JEPA-based guidance is its computational cost, since computing the JEPA score requires obtaining the singular values of the Jacobian of the JEPA encoder via SVD. The paper addresses this issue by introducing Randomized SVD to approximate the computation efficiently. The authors also empirically demonstrate that approximation errors caused by rank truncation do not significantly affect practical performance.
- Comprehensive experimental evaluation: The authors evaluate the proposed method across multiple datasets, tasks, and baseline methods, providing a thorough empirical analysis. In addition, the paper includes ablation studies examining components such as delayed guidance and the rank truncation in Randomized SVD. The authors also demonstrate that the generated samples can be useful for data augmentation, further highlighting the potential benefits of world-centric minority sampling.

Weakness
- Limited exploration of alternative JEPA models: The experiments primarily rely on DINOv2 as the JEPA model. It would be interesting to evaluate the proposed approach with other JEPA-style models (e.g., I-JEPA[1], V-JEPA[2], LeJEPA[3], or DINOv3[4]). Given the simplicity of the proposed framework, these models could likely be integrated without major modifications. Such experiments could further strengthen the empirical support for the proposed method. That said, this limitation does not significantly undermine the overall contribution of the paper.

---
[1] Mahmoud et al., “Self-Supervised Learning From Images With a Joint-Embedding Predictive Architecture”, CVPR, 2023.
[2] Adrien et al., “V-JEPA: Latent Video Prediction for Visual Representation Learning”, ICLR, 2024.
[3] Randall et al.,  “LeJEPA: Provable and Scalable Self-Supervised Learning Without the Heuristics
”, arxiv preprint, arXiv:2511.08544, 2025
[4] Oriane et al., “DINOv3”, arxiv preprint, arXiv:2508.10104, 2025

---

> ### Author Rebuttal · Authors · 2026-03-30
>
> We sincerely thank Reviewer `PUvi` for the careful reading and strong support of our work. Below we provide a response addressing your concern.
>
> ---
>
> > **W1. `PUvi` suggested evaluating the proposed approach with other JEPA encoders beyond DINOv2.**
>
> Thanks for raising this point. As per your suggestion, we conduct a new experiment incorporating DINOv3 as the guiding JEPA encoder. See the table below for details:
>
> | | CLIP $\uparrow$ | Pick $\uparrow$ | ImgR $\uparrow$ | JEPA $\downarrow$ |
> |---|:---:|:---:|:---:|:---:|
> | DDIM | **31.58** | **22.65** | **0.73** | -572.94 |
> | CADS | 31.12 | 22.35 | 0.49 | -563.17 |
> | SGMS | 31.36 | 22.57 | 0.70 | -596.93 |
> | MinorityPrompt | 31.34 | 22.58 | 0.68 | -590.25 |
> | Ours (DINOv3) | 31.51 | 22.59 | 0.71 | **-610.71** |
>
> We observe that the proposed approach works well with recent JEPA encoders like DINOv3, maintaining competitive quality and alignment while achieving significantly lower JEPA-SCORE values compared to baselines. This further highlights the flexibility and practical applicability of our framework. Also see Table 3d (in our manuscript), which demonstrates that our guidance framework also operates with MetaCLIP, a multimodal model outside the typical JEPA family, suggesting broader applicability beyond JEPA encoders.

---

> > ### Author Rebuttal · Reviewer_PUvi · 2026-04-03
> >
> > Thank you for your efforts during the rebuttal phase. I keep my positive recommendation.

---

> > > ### Author Response · Authors · 2026-04-03
> > >
> > > Thank you for acknowledging our rebuttal and for maintaining your positive recommendation. We greatly appreciate your supportive and constructive review throughout the process.

---

### Official Review · Reviewer_zDno · 2026-03-15

**Soundness:** 3
**Presentation:** 2
**Significance:** 2
**Originality:** 2
**Overall Recommendation:** 3
**Confidence:** 4

**Summary:**

The paper proposes JEPA guided diffusion for guiding with a representations of a world model, later by proposing an approximation of an earlier JEPASCORE paper based on randomized SVD and guidance they proposed a faster version of this term. Results suggest lower JEPA-SCORE with competitive sample quality and some downstream utility although the statistical signifance of the gains are minor.

**Compliance With Llm Reviewing Policy:**

Affirmed.

**Key Questions For Authors:**

Please see the previous section.

**Limitations:**

Please see the previous section.

**Strengths And Weaknesses:**

# Strenghts:
* Reasonable technical approximation to an otherwise expensive JEPAscore objective that requires full decomposition for each sampling step.
* Includes ablations over guidance strength, rank, and encoder choice.


# Weaknesses:
* In several tables, the improvements in standard quality/alignment metrics are small, and the downstream gains are reported as averages over three runs without dispersion measures. Table 4, for example, reports averages over three runs, but mean standard deviation or confidence intervals are not shown.
* Why the authors did not considered other generators? only Unet based 3 year old stable diffusions were considered, does the method also works with vit based generators too ? (e.g., flux or even sd3)
* Why are we comparing against are the authors rely heavily on JEPA-SCORE for evaluation risks metric alignment favoring the proposed method. The paper would be stronger if it included additional external metrics, especially feature-space distances or measures not derived from the JEPA model itself. one of the main metrics for measuring the quality of out of distribution generation (or minority sampling) is coverage which for the case of the SD experiments it is under performing much simpler variants.
* Since a key practical claim is that the proposed approximation is computationally preferable to direct JEPA-SCORE guidance, the comparison against that baseline should be more complete. The current computational analysis is useful, but still limited (I just found B.3 part about this) the reported runtime differences do not yet fully establish how meaningful the efficiency gain is in practice or how the tradeoff scales with settings such as rank, guidance steps, or sample count
* What will happen if we train the generators on the datasets used to train the JEPA models? More clarifications on this regard is needed. A toy example might be needed, that if we start from dataset D and train a world model (e.g., JEPA) and a generator on top of it, would it be beneficial to apply this method? isn’t it just using more information (through JEPA) to guide the generation?

---

> ### Author Rebuttal · Authors · 2026-03-30
>
> We sincerely thank Reviewer `zDno` for the insightful comments. Below, we provide detailed point-by-point responses.
>
> ---
>
> >**W1-1. `zDno` concerned that the improvements in quality/alignment metrics are small.**
>
> Please see our response to Reviewer `WVET` (W3) for details.
>
> ---
>
> >**W1-2. `zDno` pointed out that Tab. 4 does not report dispersion measures.**
>
> We now incorporate standard deviations for our method, computed over three runs with distinct seeds; see the updated table in [**this link**](https://github.com/anonymous-icml-rebuttal/tabs/blob/main/Tab4_std.png?raw=true). We omit stds for the baselines, as their values are adopted from prior work [1] that only reports averages. We observe marginal dispersion across seeds, confirming that improvements (e.g., in F1 and Recall) are stable and consistent even with smaller numbers of augmented samples (30K).
>
> ---
>
> >**W2. `zDno` questioned whether our method can work on recent generators such as FLUX.**
>
> To address this concern, we conduct new experiments on FLUX.1-Schnell; see the table in [**this link**](https://github.com/anonymous-icml-rebuttal/tabs/blob/main/Tab_flux.png?raw=true) for details. Our method demonstrates consistent gain even under the MMDiT-based architecture with a distinct parameterization type. Also note that the reported JEPA-SCORE values are computed with DINOv3, distinct from the guidance encoder (DINOv2), addressing the concern of metric alignment (W3).
>
> ---
>
> >**W3-1. `zDno` concerned that relying on JEPA-SCORE for evaluation risks metric alignment favoring our approach.**
>
> The reviewer is kindly reminded that our task is explicitly defined as generating samples that are rare under the JEPA-induced prior (i.e., yielding low JEPA-SCORE values), which captures broader semantics than typical generative priors (Sec. 4.1 and Fig. 2). Evaluating with JEPA-SCORE is therefore not circular but a direct measure of task success. Note that JEPA-SCORE is not a metric proposed by us, but an independently established density measure [2]. In fact, this evaluation approach is analogous to prior minority sampling works (MG, SGMS, BnS), which employ an existing density metric (e.g., AvgkNN) to both define and evaluate minorities.
>
> Nonetheless, we conduct cross-evaluation where JEPA-SCORE is measured with DINOv3, independent of the guidance encoder (DINOv2), updating our main results (Tabs 1 and 2) accordingly; see [**this link**](https://github.com/anonymous-icml-rebuttal/tabs/blob/main/Tabs12_DINOv3_JS.png?raw=true). Note that our method achieves consistently lower JEPA-SCORE even under an independent evaluation model. See also W2 for cross-evaluation on FLUX.
>
> ---
>
> >**W3-2. `zDno` raised concerns that our approach underperforms some baselines in coverage on the SD results.**
>
> We kindly note that reference-based metrics like coverage are less reliable in T2I generation due to the inherent domain gap between reference data (e.g., MS-COCO) and generated samples [3]. This is evident from the inferior coverage of other minority baselines (SGMS, MinorityPrompt) compared to DDIM, indicating this is not specific to ours. In settings where these metrics are reliable (CelebA, ImageNet-256), our method achieves the best coverage among all baselines (Tab. 1). This is further supported by our user study (Tab. 7) where users prefer our samples in atypicality (as well as quality).
>
> ---
>
> >**W4. `zDno` suggested a more complete cost analysis, including a comparison with exact JEPA-SCORE guidance and the runtime scaling with key design choices.**
>
> Our method requires 10.44 seconds per sample (Tab. 6), while exact JS guidance requires **280** seconds, approximately **27 times** slower, validating the practical necessity of our approximation. We also provide runtime scaling with respect to rank k, guidance interval N, and deferral ratio tau; see [**this link**](https://github.com/anonymous-icml-rebuttal/tabs/blob/main/Tabs_costs.png?raw=true). Our parameter choice (k, N, tau) = (9, 3, 0.8) strikes a favorable balance between cost and performance.
>
> ---
>
> >**W5. `zDno` questioned whether our approach remains beneficial when the JEPA encoder and the generator share the same training data.**
>
> In such cases, generator-centric and world-centric minorities would coincide, and our method would have a similar effect to conventional minority samplers. However, in practice, generators are often trained on smaller-scale data (e.g., CIFAR-10 with 50K images), whereas JEPAs are trained on large-scale diverse data (e.g., DINOv2 on LVD-142M with 142M images), capturing a broader prior that better approximates the real-world distribution. It is this gap that our approach leverages; see Fig. 2 and Sec. B.1.
>
> ---
>
> **References**
>
> [1] Boost-and-Skip: A Simple Guidance-Free Diffusion for Minority Generation, ICML 2025.
>
> [2] Gaussian Embeddings: How JEPAs Secretly Learn Your Data Density, arXiv 2025.
>
> [3] SDXL: Improving Latent Diffusion Models for High-Resolution Image Synthesis, ICLR 2024.

---

> > ### Author Rebuttal · Reviewer_zDno · 2026-04-03
> >
> > Thanks the authors for the detailed rebuttal and additional experiments.
> >
> > W2., W3.1 Don't they share a large number of training sampels? (i.e., between v2 and v3)?

---

> > > ### Author Response · Authors · 2026-04-03
> > >
> > > We thank the reviewer for the follow-up question.
> > >
> > > **Regarding data overlap between DINOv2 and DINOv3,** we would not characterize the two training sets as largely overlapping. The two models draw from fundamentally different source pools: DINOv2's training data (LVD-142M) is curated from a 1.2B web-crawled image pool, whereas DINOv3's training data is primarily curated from a 17B pool of public Instagram images. While both pipelines incorporate common public datasets such as ImageNet-1k/22k, these constitute less than 1% of DINOv3's training set (over 1.6B images).
> > >
> > > **That said, even if some large overlap exists,** this does not undermine our cross-evaluation. DINOv3 is trained on a substantially larger dataset with a different architecture and training recipe, capturing a broader visual prior than DINOv2. The fact that our method (guided by DINOv2) achieves consistently lower JEPA-SCORE even when evaluated under this broader model suggests that the generated samples are genuinely atypical under a wider prior, rather than being artifacts of overfitting to DINOv2's specific density landscape.

---

### Decision · Program_Chairs · 2026-04-30

**Decision:**

Accept (regular)

**Comment:**

This paper proposes a JEPA-guided diffusion method for minority sampling. It introduces the JEPA-score to estimate real-world density and an efficient approximation for practical use. The authors respond well to concerns about data overlap between DINOv2 and DINOv3. They clarify that the datasets are largely different and that DINOv3 is broader and larger.